# International wildlife trade quotas are characterized by high compliance and coverage but insufficient adaptive management

Oscar Morton [1,2] ✉, Vincent Nijman[3] & David P. Edwards [2]

Effective management of international wildlife trade is crucial to ensure sustainability. Quotas are a common trade management tool and specify an annual number of individuals to be exported, yet at present there is no global assessment of quota coverage and compliance. Using over 7,000 country–year specific reptile quotas established under the Convention on International Trade in Endangered Species of Wild Fauna and Flora (CITES) covering 343 species, we quantify quota coverage, compliance, trade trends pre-quota and post-quota setting and whether quotas likely represent adaptive management. Quotas predominantly concerned live wild-sourced reptiles, with only 6.6% of live non-zero quotas exceeded and 4.5% of zero quotas subverted. For 62.3% of species, quotas were established higher than pre-quota trade, with traded volumes post-quota mainly unchanged or higher than pre-quota establishment, thus potentially facilitating sustainable trade. Over 38% of quota series of species remained at the same level each year, with the longest-running quotas proportionately updated the least, indicating that many quotas do not change adaptively in response to changing threats to species through time. Greater specificity in exactly what quotas cover, justification for unchanged quotas and transparency over quota determination are needed to ensure that high compliance equates to sustainable use.

Effective management of wildlife trade is crucial in ensuring sustainable trade, supporting local livelihoods, conserving species and ensuring food security[1,2]. At least 24% of terrestrial vertebrates and more invertebrates and plants, are traded in some form with over 12,000 species being recently internationally traded[3–6]. While well-managed trade can bolster conservation efforts and populations, poorly managed trade can correlate with severe declines in species abundance[7].

The Convention on International Trade in Endangered Species of Wild Fauna and Flora (CITES) regulates international trade in CITES-listed species. Parties implement the Convention domestically to ensure that trade causes no detriment to species survival. Many do so using quotas, prespecified volumes of individuals (or derived products) that can be exported by the Party over a 12 month period. Since 1997, over 25,000 quotas have been set by Parties. Although not explicitly mandated in the Convention, the Conference of Parties (CoP) emphasizes quotas as essential management tools (CITES[8]), stating that national quotas should be used to ensure that "the species is maintained throughout its range at a level consistent with its role in the ecosystem" (CITES[8]).

[1]Ecology and Evolutionary Biology, School of Biosciences, University of Sheffield, Sheffield, UK. [2]Department of Plant Sciences and Conservation Research Institute, University of Cambridge, Cambridge, UK. [3]Oxford Wildlife Trade Research Group, Oxford Brookes University, Oxford, UK. ✉e-mail: om403@cam.ac.uk

As such, scientifically informed annual quotas can take the place of several separate non-detriment findings (NDFs; scientific assessments to determine whether trade would have a detrimental effect on species persistence). Quotas can also be set directly for limited reasons, including by the CoP to manage trade when species are moved from Appendix I to II (ref. 9) or as part of the Review of Significant Trade process[9].

Building mechanisms and tools to deliver sustainable trade (for example, via quotas) is critical and can avoid trade shifting to potentially more damaging illicit pathways[10]. In Pakistan, community-based trophy hunting quotas incentivized and funded the conservation of species declining due to poaching (Himalayan ibex *Capra ibex sibirica*, Blue sheep *Pseudois nayaur* and Astore markhor *Capra falconeri falconeri*)[11]. Zero quotas can also be used to halt unsustainable trade, although the effectiveness of bans[12] and their negative consequences[13] remains contested.

Ensuring that export quotas positively contribute to sustainable trade requires three things: (1) quotas represent scientifically informed sustainable offtakes; (2) quotas are complied with by national agencies implementing CITES; and (3) quotas are adaptive, adjusting through time to reflect new information. These assumptions have not been tested at scale. While adaptive management has long been integral to fisheries management[14] and is regularly advocated in CITES policy[15], how it would fit within the current CITES framework in unclear[16]. Others[17] probed the first and third assumptions for leopard (*Panthera pardus*) quotas, concluding that they were based on flawed ecosystem understanding, at odds with adaptive management, set unjustifiably high and thus unsustainable in certain areas (for example, Limpopo, South Africa).

Many studies also suggest instances of quota non-compliance[18,19]. In the Indonesian blood python (*Python brongersmai*) trade, when wild-taken quotas are filled, additional individuals are falsely reported as captive-born[20] or stockpiled and illegally smuggled[21]. Similarly, ball python (*Python regius*) exports from Togo have frequently breached quotas by many thousands of individuals[19].

Without a sound understanding of quota coverage, compliance and adaptiveness globally, the effectiveness and importance of quotas as a management tool cannot be known. We tackle this through a comprehensive study of all CITES-listed reptile export quotas since 1997. Unlike certain listed taxa (for example, birds), the international legal trade in listed reptiles trade remains vast (>3 million individuals annually), greater than all other listed vertebrate trade combined[22,23] and has the most quotas of any taxon. Effective reptile quota management is thus critical for international trade sustainability. We have four objectives: (1) assess quota non-compliance; (2) quantify volume changes for pre-quota to post-quota setting; (3) determine whether quotas are consistently updated as per adaptive management; and (4) identify key species and country combinations not using quotas.

## Results

### Quota coverage
After deduplication, cleaning and categorization, we found 7,761 reptile quotas set between 1997 and 2023 (each specifying the exporter, year and taxa) from 70 Parties; most also specified the term monitored (for example, live and skins) and the source (for example, wild or captive). There were 4 subspecies-level, 333 species-level, 5 genera-level and 2 family-level quotas (Supplementary Fig. 1). Most quotas concerned wild-sourced trade (74.7%, Supplementary Fig. 2). Almost 40% of quotas did not specify the type of term managed and those that did focus predominantly on live individuals (43.5%, Supplementary Fig. 2). Combinations of incomparable terms that cannot feasibly be converted to whole animals occurred in a small number of quotas, for example, small leather pieces and skins or skins and meat (Supplementary Fig. 3).

### Quota compliance
We evaluated compliance by focusing on the 2,846 live reptile quotas that could be verified against reported trade. Between 1997 and 2021,

134 zero quotas were issued for live reptiles covering 29 species from six CITES Parties, with Malaysia setting 91 (Fig. 1a). In 95.5% of cases, zero quotas were complied with (Fig. 1c,e). Zero-quota breaches (six) originated from Malaysia (four), Madagascar (one) and Niger (one), with breaches ranging from eight painted terrapin (*Batagur borneoensis*) exported from Malaysia to Japan in 2018, to 532 Geyr's spiny-tailed lizards (*Uromastyx geyri*) exported from Niger to unknown importers in 2009.

The remaining 2,712 quotas set for live reptiles (1997–2021) varied drastically in size: the largest wild-sourced quotas concerned Colombian and Costa Rican green iguana (*Iguana iguana*) exports and the largest captive-sourced quotas also covered Colombian green iguana and Siamese crocodile (*Crocodylus siamensis*) from Vietnam (Supplementary Fig. 5). Most quotas were set by Indonesia (567), Madagascar (563), Ethiopia (272), Guyana (262) and Suriname (193, Fig. 1b). Known quota breaches occurred in 6.3% of quotas, with Madagascar having the most instances (82 instances, 14% of their set quotas), followed by Ghana (24 instances, 23% of their set quotas) and Indonesia (21 instances, 4% of their set quotas) (Fig. 1d,f). While trade was generally compliant, the distribution of percentages of quota allowances used is highly right-skewed with a few substantial breaches (Fig. 2g). The greatest quota breach was for 1,104.8% of the Madagascan quota for wild-taken lined flat-tail gecko (*Uroplatus lineatus*) in 2010 (696 exported-reported and 318 importer-reported individuals against a quota of 63), with trade in other years ranging between 0% and 120.0% of the allocated quota being traded (2005–2021). On average, only 49.8% of the maximum permitted offtake under quotas was traded (for example, only 498 individuals traded out of a quota for 1,000); this increased to 69.0% after removing 717 quotas for which there was no reported trade (Fig. 1g).

Examination of instances where exporter-reported volumes are quota compliant but importer-reported are not, highlight 35 species across 63 quotas where quotas were also probably breached (Table 1 and Supplementary Table 2). Breaches ranged from 0.7% to 660% in excess of quota volumes, with the largest quota breaches concerning captive-bred ball python and wild-sourced Home's hinged-backed tortoise (*Kinixys homeana*) and Senegal chameleon (*Chamaeleo senegalensis*) all exported from Ghana.

### Pre-quota to post-quota trade patterns
Using 69 national-level quota time series that had ≥5 consecutive years of quotas and ≥5 years of pre-quota and post-quota trade records, we find on average that quotas were set a median of 0.82 s.d. (90% HDI 0.45–1.12, probability of direction (pd) = 100.00%) higher than pre-quota volumes and 0.88 s.d. (90% HDI 0.72–1.05, pd = 100.00%) higher than traded post-quota volumes. On average, pre-quota traded volumes steadily increased up to the year in which quotas were implemented before plateauing at comparable levels to pre-quota trade (median = −0.06, 90% HDI −0.34–0.24, pd = 61.95%; Fig. 2a and Supplementary Table 3).

Over 46% (32 of 69) of quota levels were set above pre-quota trade volumes with no subsequent temporal change over time (step change panel 2; Fig. 2b). This increases to 62.3% (43 of 69) of quota series when all quotas representing a step increase and a temporal change are considered ('step change' plus relevant 'step and trend change' panels 2, 11, 12, 14, 15, 16 and 18; Fig. 2b and Supplementary Fig. 6). This includes some instances (for example, three Madagascan *Brookesia* chameleon species) where there was no pre-quota trade under CITES with trade beginning post-quota establishment, potentially demonstrating the effective use of quotas to manage trade from the outset. The second most common relationship was increasing pre-quota volumes with quota levels then set lower and maintained consistently low (14.5%, 10 of 69), demonstrating that quotas are both effectively used to promote managed trade and to curb increasing (potentially concerning or unsustainable) trade. Quotas were never set lower than pre-quota volumes when trade was declining or stable through time.

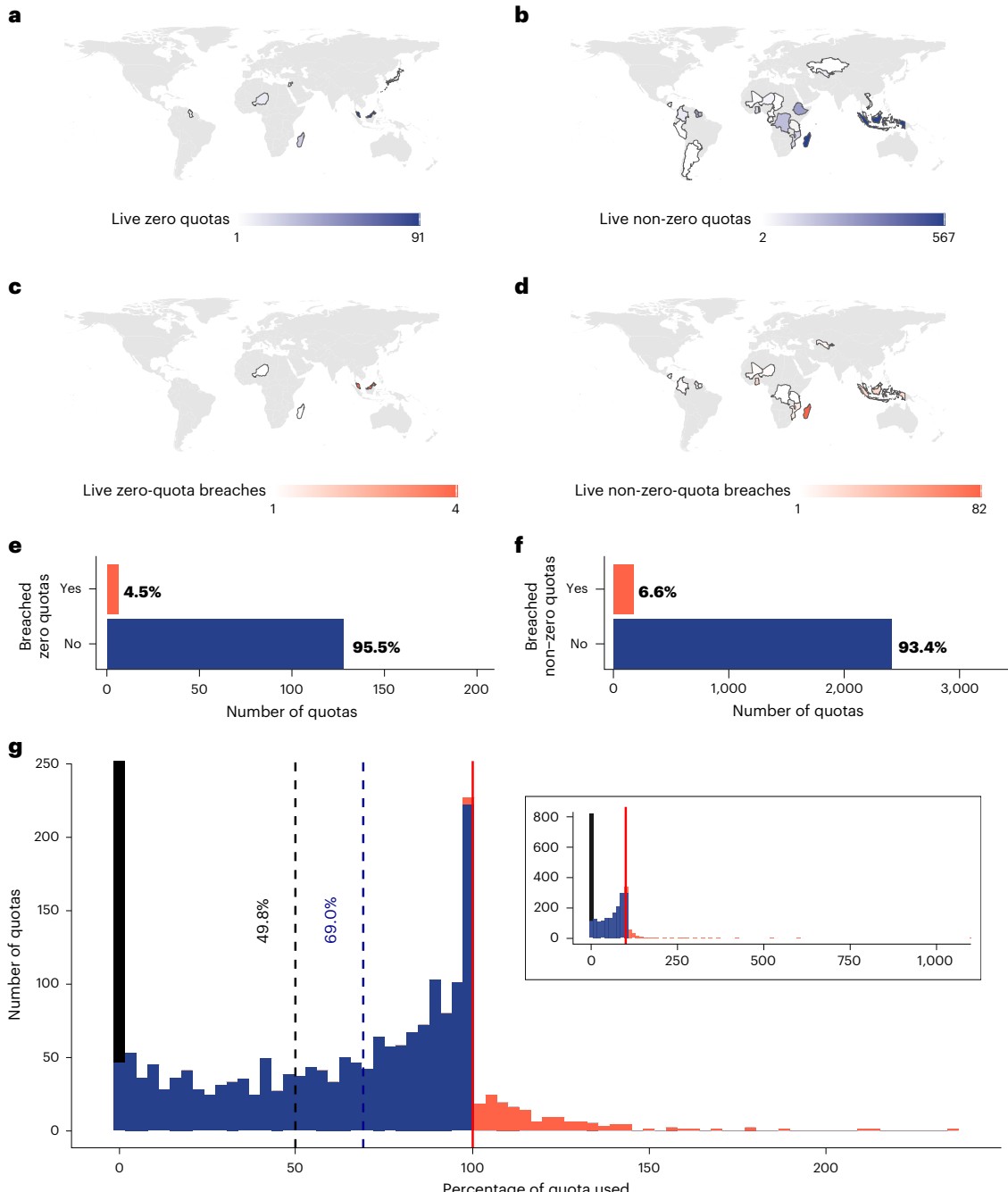

**Fig. 1 | Live reptile quota compliance 1997–2021. a,b,** Counts of total live zero quotas (**a**) and non-zero trade quotas (**b**). **c,d,** Counts of live zero-quota breaches (**c**) and non-zero-quota breaches (**d**). **e,f,** Tallies of the total number of adhered to and breached zero quotas (**e**) and non-zero quotas (**f**). **g,** Percentage of each quota used (0% indicates the species had a quota but was not traded, 100% would indicate exactly the quota amount was traded and percentages above 100% indicate quota breaches). The x axis of the main graph is truncated at 250% quota use and the y axis at 250 quotas for clarity, the full distribution is shown in the inset. The dashed black line indicates the average percentage of a quota used, including species never traded but under quota (including the 0% values), dashed blue line indicates the average percentage of a quota used for species that have been traded (excluding the 0% values) and the solid red line indicates 100% quota use. Only 20 quotas had a percentage use over 250%.

Similar to quota levels, traded volumes post-quota setting were most commonly higher (with no further change) than pre-quota volumes (step change panel 2; 30.4%, 21 of 69, Fig. 2c), indicating instances where quotas may have been used to promote trade. The second most common relationship was no change in pre-quota and post-quota volumes through time (panel 1, 17.4%, 12 of 69), often due to no trade pre-quota or post-quota despite high quotas (for example *Pelusios* sp. from Mozambique). In 40.1% (panels 5, 6, 7, 12, 13, 15, 17 and 18, 28 of 69, Supplementary Fig. 6f) of cases, post-quota

volume trends declined relative to pre-quota trends, most often due to increasing or flat pre-quota volume trends and flat or decreasing post-quota volumes (Fig. 2c). For example, Niger's trade in Geyr's spiny-tailed lizard rapidly increased pre-quota but post-quota volumes declined to zero in line with quotas for the species. In over a fifth (15 of 69) of cases, actual volumes that were increasing pre-quota fell and remained either temporally constant (panel 13, 14.5%, 10 of 69) or temporally decreasing (panel 17, 7.2%, 5 of 69, Fig. 2c) after quotas were implemented.

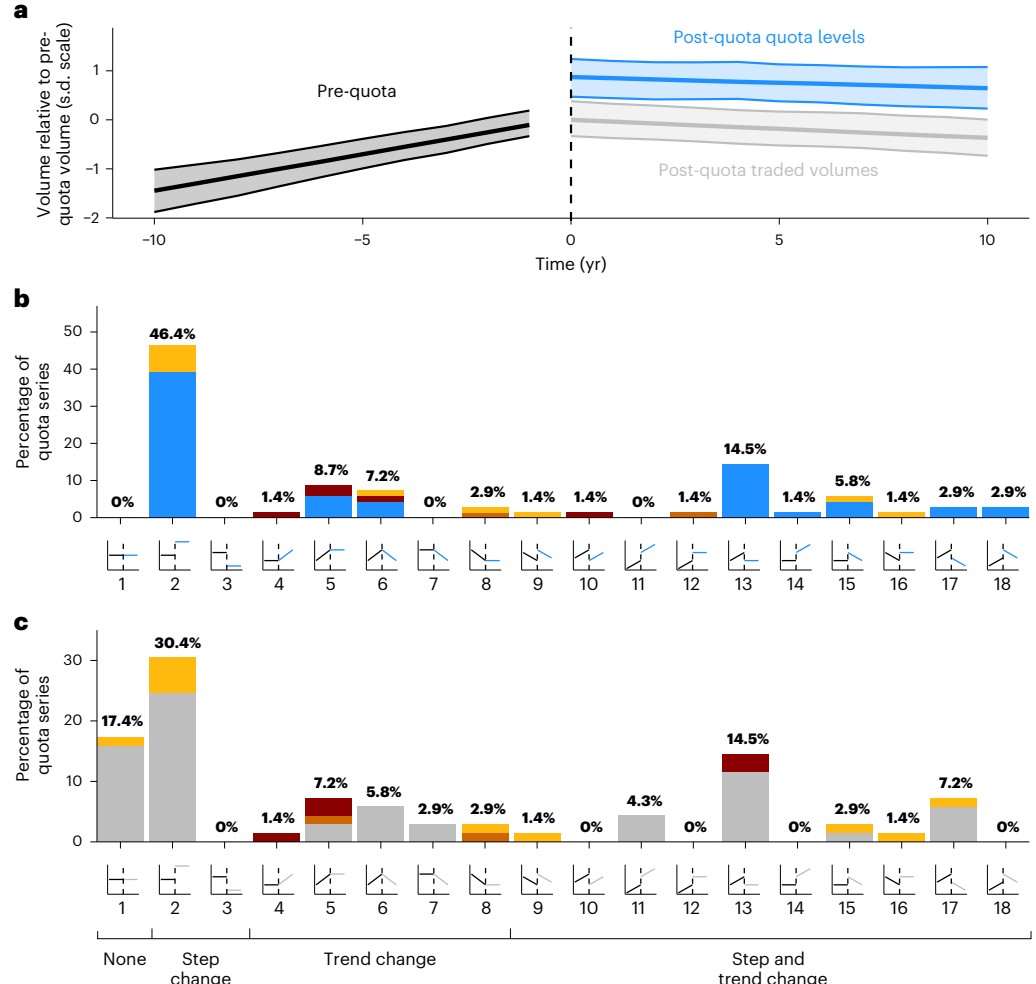

**Fig. 2 | Trade volume pre-quota to post-quota setting. a**, Volumes trends 10 years pre-quota and post-quota implementation for the average species exporter. Volumes are on the species-exporter standardized scale where 1 denotes a 1 s.d. higher volume relative to the traded volume the year before quotas were first issued. Lines denote the median and intervals the 90% highest density interval (HDI). **b**, Summary of quota level associations with pre-quota trade volumes. **c**, Summary of post-quota actual traded volumes association with pre-quota trade volumes. Bars show the percentage of quota series exhibiting that relationship. Bar colour denotes the species threat status: dark red (assessed as globally threatened as per the IUCN Red List 2023 and probably threatened

by international trade), orange (not globally threatened as per the IUCN Red List 2023 and likely threatened by international trade), yellow (globally threatened as per the IUCN Red List 2023 and not likely threatened by international trade), blue and grey bars are for taxa that are neither globally threatened nor probably threatened by trade. For both **b** and **c**, the *x* axis concept figures show the pre-quota to post-quota quota level or traded volume relationship with pre-quota volumes and are numbered for reference. See Supplementary Table 5 for a breakdown of relationships and how they are classified and Supplementary Figs. 6, 7 and 8 for species-level plots of post-quota levels and traded volumes, respectively.

## Are quotas managed adaptively?

In total, there were 624 species, term, source, purpose and exporter-specific quota time series from 2 to 27 years in length (Fig. 3). Only 64 (10.3%) quota series volumes changed yearly, most concerning short 2–5 year series (Fig. 3a). Conversely, 240 (38.5%) series concerned volumes that never changed, including 41 quotas running for ≥10 years (Fig. 3b). Of 383 long-term series (≥5 years), 62.4% (239) changed less than once every 5 years.

A change-point analysis suggests that longer quotas stagnate, receiving fewer changes relative to their length (Fig. 3c). Before the estimated change point, changes increased with each extra year of quota length, albeit only at a rate of one additional change every 3.4 years (median increase 0.29, 90% CI 0.25–0.32, pd = 100.00%). After the change point, there was no association between length and changes (median increase −0.01, 90% CI −0.11–0.10, pd = 56.90%), a clear change from the pre-change association (Supplementary Table 4). Stagnation occurred after 17.4 years (90% CI 16.4–18.8). For example, the average 15 year quota is changed four times (median estimate), whereas a 25 year quota is changed 4.6 times, representing a drop rate of change from 0.27 to 0.18 per year.

## Gaps and opportunities in quota coverage

Of the 8,773,121 live, wild-sourced reptiles traded (1997–2021), 4,416,476 were traded under quotas. The remaining volume (4,356,645 individuals across 338 species and 111 exporters) of species traded without quota management included over 2 million data deficient or not evaluated (by the IUCN) individuals across 248 species by 81 exporters (Fig. 4a–c). Additionally, species not covered by quotas included 944,366 globally threatened (vulnerable, endangered or critically endangered as per the IUCN Red List at the time of trade) individuals from 65 species (traded by 70 exporters, Fig. 4d–f); and 572,406 individuals from 49 species (traded by 43 exporters) identified as being likely threatened by international trade (Fig. 4g–i). In recent years, the number of species-exporter combinations of these species of conservation concern has fallen consistently (Fig. 4d,g). Trade volumes lack such a clear decline, primarily because of voluminous trade

**Table 1 | The ten highest potential breaches where exporter-reported volumes were compliant but importer-reported volumes breached quotas**

| Taxon | Current IUCN status | Party | Year | Quota | Exporter reported (volume (%)) | Importer reported (volume (%)) |
|---|---|---|---|---|---|---|
| *Uromastyx geyri* | Near threatened | NE | 2008 | 0 | 0 (–) | 200 (–) |
| *Brookesia minima* | Endangered | MG | 2017 | 0 | 0 (–) | 21 (–) |
| *Python regius* | Near threatened | GH | 2009 | 200 | 140 (70%) | 1320 (660%) |
| *Python regius* | Near threatened | GH | 2008 | 200 | 130 (65%) | 960 (480%) |
| *Python regius* | Near threatened | GH | 2011 | 200 | 200 (100%) | 770 (385%) |
| *Kinixys homeana* | Critically endangered | GH | 2002 | 340 | 119 (35%) | 1109 (326%) |
| *Chamaeleo senegalensis* | Least concern | GH | 2002 | 1,500 | 1222 (82%) | 3346 (223%) |
| *Furcifer lateralis* | Least concern | MG | 1999 | 2,000 | 1806 (90%) | 4398 (220%) |
| *Python regius* | Near threatened | GH | 2010 | 200 | 50 (25%) | 420 (210%) |
| *Kinixys homeana* | Critically endangered | GH | 2001 | 340 | 0 (0%) | 683 (201%) |

Percentages shown are percentages of the set quota reported as traded in that year. Party codes: NE, Niger; MG, Madagascar; GH, Ghana.

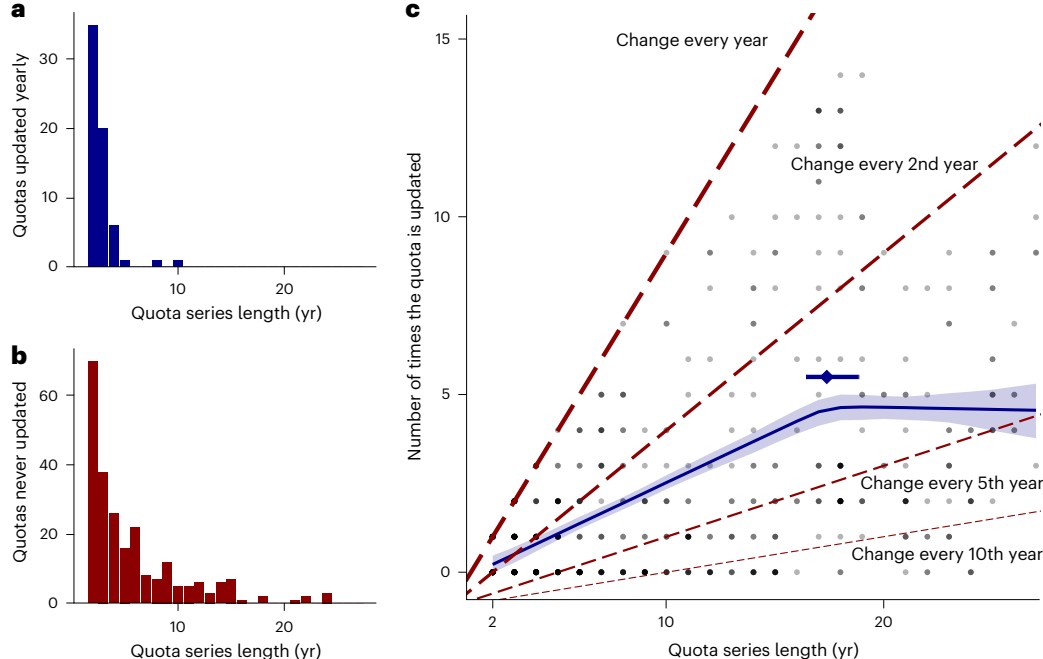

**Fig. 3 | Frequency of quota changes. a**, Tally of quotas of each length that changed each year they were set. **b**, Tally of quotas that never changed for each length. The **a** and **b** x axes are both scaled to 27 years, the maximum possible quota series length (1997–2023). **c**, Number of quota updates across quota lengths (across 624 quota time series at least 2 years in length). Diagonal reference lines show the expected distribution of quotas if they were unique each year (for example, a quota running for 10 years would have been updated nine times), changed on average every 2 years, changed on average every 5 years and changed on average every 10 years. The blue line denotes the median posterior expectation of the change-point model, the shaded band the 90% HDI, the diamond point and error bar above this show the median change point and the 90% HDI around this estimate. Raw data points are translucent to aid visualizing overlaid points.

in the vulnerable (by IUCN) and likely threatened by international trade alligator snapping turtle (*Macrochelys temminckii*).

In recent years (since 2016) the diversity of species and Parties trading species outside of quotas has drastically shrunk relative to the historical occurrences. The dominant Parties recently trading not evaluated, data deficient, globally threatened or species likely threatened by international trade include Suriname, Ghana, Togo, Indonesia and Sudan (Fig. 4). There have been only 14 species-exporter combinations (11 species from 8 exporters) where species likely threatened by international trade have been traded and not quota managed (annual species-exporter volumes range from 1 to 43,718 individuals, Supplementary Table 6). The most voluminous is alligator snapping turtle exports from the United States (29,801–43,718), with the species recognized as being threatened by international trade and the US Fish

and Wildlife Service note that both legal and illegal harvest contribute to this threat. Similar species likely threatened by international trade that lacked quotas include the infrequently traded and endangered pig-nosed turtle (*Carettochelys insculpta*) exported from Indonesia and vulnerable Senegal flapshell turtle (*Cyclanorbis senegalensis*) from the Democratic Republic of Congo, Ghana and Togo. Togo recently began issuing quotas for the species (2019–2023), with the quotas (50–200 live wild-taken individuals annually) higher than previous traded volumes (83 in 2017; 9 in 2018), while most recent trade outside quotas originates from Ghana with 235 live wild-taken individuals traded.

## Discussion

Our analysis highlights large-scale quota compliance and quantifies the diversity of pathways in which quotas can promote trade in species,

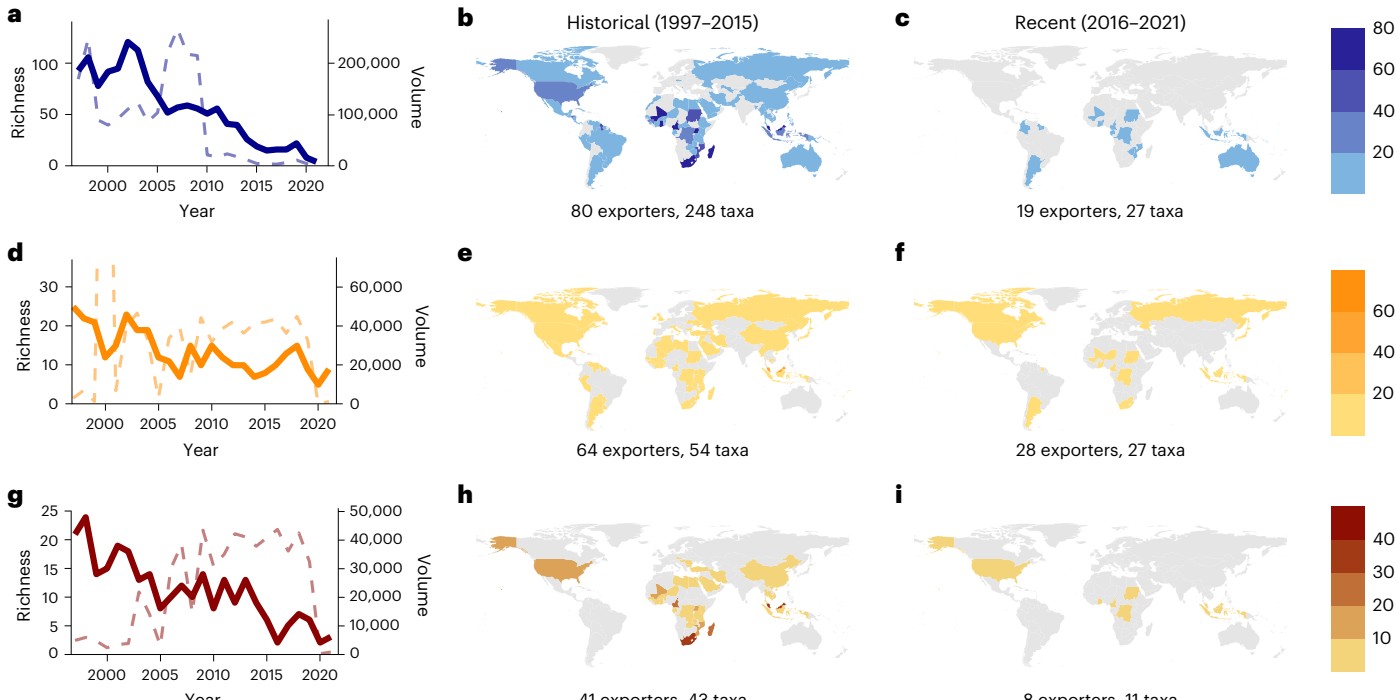

**Fig. 4 | Global shifts in trade not managed by national export quotas.**
**a**, Trends in traded species richness and volume for species classed as data deficient or not assessed by the IUCN not managed with national export quotas (blue tones). **b,c**, Traded species richness for species classed as data deficient or not assessed by the IUCN not managed with national export quotas from 1997–2015 (**b**) and from 2016–2021 (**c**). **d**, Trends in traded species richness and volume for species classed as threatened by the IUCN (vulnerable, endangered, critically endangered) not managed with national export quotas (orange tones). **e,f**, Traded species richness for species classed as threatened by the IUCN (vulnerable, endangered, critically endangered) not managed with national export quotas from 1997–2015 (**e**) and from 2016–2021 (**f**). **g**, Trends in traded species richness and volume for species assessed as likely threatened by international trade not managed with national export quotas (red tones). **h,i**, Traded species richness for species assessed as likely threatened by international trade not managed with national export quotas from 1997–2015 (**h**) and from 2016–2021 (**i**). In **a**, **d** and **g** the dashed lines show the volume of these species traded each year and correspond to the right *y* axis.

prevent increasing trade and reduce trade below previous levels. Similarly, these results highlight the general decline in the number of poorly understood (data deficient or not evaluated by the IUCN), globally threatened and species likely to be specifically threatened by international trade traded outside of quotas. Conversely, the analysis also highlights a lack of adaptive management for many international export quotas published for CITES-listed reptiles. While quotas have great potential as a key management tool, we urgently need to improve their relevance and transparency to ensure international trade is sustainable for such a heavily traded taxon.

## Quotas as a tool for ensuring sustainable trade

To be effective, quotas must be complied with. Our results demonstrate this is largely the case but also highlights that a proportion of quotas are ambiguously worded, creating a barrier to effective monitoring of compliance (Supplementary Fig. 3). For example, Indonesia has set ambiguous quotas covering both 'skins and skin products' for 15 species, including large quotas for *Crocodylus novaeguineae*, *Varanus salvator* and *Naja sputatrix* as recently as 2020. For quotas by some Parties, this may be an historical issue, for example Nicaraguan spectacled caiman (*Caiman crocodilus fuscus*) quotas aggregated incomparable terms up to 2010 (skins, sides, tails, bellies, bodies and leather products) but have since changed this to use comparable terms. Yet the general trend through time appears to suggest increasing numbers of incomparable terms in use rather than less (Supplementary Fig. 3). Likewise, Indonesia exports tens of thousands of live oriental rat snakes (*Ptyas mucosa*) annually, despite national quotas stipulating 334–500 live individuals annually (between 2006 and 2023). This occurs because of a separate annual quota for up to 90,000 'skins' or 'skins and meat', most of which is traded live and assumed to be destined for processing into skins. While such instances represent a quota-compliant number of individuals in trade, they contribute to the opaqueness of quota regulation, hindering transparent compliance assessments.

Using quotas to streamline exporter sustainability assessments by aggregating several NDFs requires robust practices that can be scaled up to a Party's annual export rather than just a population. Yet, there seems little evidence that NDF practices effectively assess offtake sustainability in the first place[23,24]. A review of accessible NDFs (totalling 36) by the CITES Secretariat highlighted only 41.7% fully considered population trends, 44.4% incorporated other threats and 36.1% fully considered how the precautionary principle would apply[24]. For instance, Thailand made positive NDFs for several seahorse species but subsequently failed to support these with evidence[25]. Independent assessments for two species concluded a negative NDF for one and changes to current practices for the other[25,26]. If NDFs are not accurately assessing sustainable offtakes, then inappropriately set quotas just compound this issue.

A harder issue to address is that quotas are explicitly for export, not harvest. Such nuance is important because multiple years' worth of individuals can be unsustainably harvested in a single year and stockpiled for quota-compliant export over subsequent years[20,21,27,28]. Exporter corruption is another cryptic issue, enabling the reporting of quota-compliant volumes, yet the importer reports volumes in apparent breach of national quotas (Supplementary Table 2). Others[29] report several years where Indonesian exports of South Asian box turtle (*Cuora amboinensis*) potentially breached their quotas when

importer-reported volumes are considered. Our study reveals this issue is more widespread, finding a further 34 species (totalling 80,904 individuals) with importer-reported volumes that breach national quotas.

## Quota compliance or quota irrelevance

For many species (for example, *Varanus beccarii, V. dumerilii, V. jobiensis* and *V. rudicolis* exports from Indonesia), quotas were used to curb high and rising export volumes. However, quotas were most commonly set higher than previous or current trade volumes and often went many years without change. This is not inherently a concern where quotas are merely higher than current patterns of demand and are sustainable. For example, Mozambique's African forest turtle (*Pelusios gabonensis*) quotas were consistently set at 10,000 live individuals (1999–2008) despite no reported trade during that time (and only 93 individuals reported by importers). In total, there are 58 non-zero-quota time series (species, exporter, source and purpose series) for live individuals where no trade has been reported. Setting quotas higher than previously recorded trade puts extra emphasis on the need for valid sustainability assessments that can rigorously justify elevated volumes.

Similarly, quotas that remain unchanged through time could be sustainable but given rapid and severe environmental change (through deforestation, illegal hunting or climate change), Parties must justify the relevance of historic quota levels with up-to-date data, especially for species threatened by several stressors[30]. This specific issue is identified in CITES Resolution Conf. 14.7 (Rev. CoP15) where the concern that quotas may not be updated to reflect changing climatic threats such as droughts is highlighted[8]. Alternative options include replacing fixed quotas with adaptive quotas directly linked to monitoring of species populations like South Africa's recent shift from fixed trophy quotas to 0.5% of the population for black rhinoceros (*Diceros bicornis*) exports[31], maintaining socioeconomic benefits and population integrity[32].

Across many taxa and over time, there is a clear concern that quotas are not always based on robust data or set sustainably[33,34]. Instead, some authors suggest that quotas can be based entirely on subjective information (concerning *Aquilaria* quotas[35]), guesswork rather than hard data (concerning Indonesia's quotas[36]) or at odds with the principles of sustainable use (for leopard quotas[17]). While our results highlight quotas are largely complied with, have increasing coverage of threatened species and can address rising trade volumes, our results also highlight potential issues with their use. These issues may present a barrier to sustainability, such as incomparable terms being combined in a single quota and quotas remaining unchanged for many years, reflecting a lack of adaptive management. Liaising with specific Parties and species could offer insight into quota setting practices, sustainability and any reasons for breaches but at a global scale for hundreds of species-Party combinations this will quickly become unmanageable. Open and transparent methods and justification for quotas are needed to ensure sustainability at scale.

**Conclusions and recommendations.** We echo several previous calls for serious international discussion and potential reforms to CITES to more robustly embed an evidence-based framework and justified quotas in national practice[23,34]. Legally trading wildlife internationally does not equate to sustainable international wildlife trade by default[17,33,37], there must be evidence showing this and this must be publicly available.

Systematic policy changes up to and including amendments to the Convention are needed to proactively support the evidenced sustainability of trade. We suggest that the following amendments would improve assessing compliance and confidence in sustainable offtake. First, establishing and enforcing a minimum acceptable level of information (for example, one or more comparable terms) as potentially ambiguous quotas only further entrench uncertainty, this would require a minor revision to the Annex Guidelines of Resolution Conf. 14.7 (Rev CoP15). Second, requiring Parties to submit supporting NDFs

for quotas each time they are communicated to the CITES Secretariat, thus allowing scrutiny of the quota's validity and relevance to current populations and threats. The Global Biodiversity Framework 2030 Target 5 focuses on the sustainable use and trade of wild species yet has no listed data indicators to capture the legal wildlife trade (excluding fish stocks), creating such data must be a priority. Such an amendment would be substantial, and even if accepted by most Parties, slow to enter into force but by identifying capacity and implementation gaps in Parties NDF practices, such NDF reporting could be used to leverage increased funding for key Parties from global sources (for example, the Global Biodiversity Framework Fund). Sustainable international trade cannot exist without an evidence-based framework and such evidence is currently sorely lacking.

## Methods

### Data preparation

All reptile quotas for CITES-listed species were downloaded from https://speciesplus.net/ (accessed 27 June 2023), totalling 8,455 rows (uniquely recorded quotas), spanning 1997–2023. Duplicated rows were removed (where all data fields were identical), reducing this to 8,445 rows. Quotas are stored in a verbose manner with all details of what type of trade the quota concerns recorded as a text string under 'notes'. All the distinct notes were extracted (*n* = 690) and manually interpreted to yield the term (for example, live, skins or trophies), source (for example, wild, F1 captive and ranched) and purpose (for example, all, commercial or hunting) of trade detailed in the note. As per Conf. 14.7 (Rev CoP15) Annex 1, Paragraph 11, if the source is not specified we interpret the quota as referring to wild-sourced trade (CITES[8]). For example, a quota with a description simply as NA (no data entered) would be interpreted as covering wild-taken trade, for any purpose and any term, likewise a quota description of 'live' would be interpreted as covering all wild-taken, trade in live individuals for any purpose. A further potentially ambiguous case arises around the use of 'all'. Often this was paired with more detail, for example, 'all, wild-taken' or 'all, wild, ranched and captive-bred' in which case the quota was assumed to cover all trade terms and all trade purposes from the sources specified. In the few cases for which simply 'all' was specified we assumed this to cover trade of all terms, from all sources for all purposes. Likewise, 'all, live' would be assumed to cover the trade of live individuals, from all sources for all purposes.

Reptiles had genus-level, species-level and subspecies-level quotas. We detected four instances where the rank was specified at a species-level, yet the taxon given was a subspecies. In all cases the subspecies were not explicitly CITES-listed, with only the species listed and ever recorded in trade. In these instances, we resolved the subspecies to the species-level. There were several instances where updated taxonomic changes have been adopted at the CoP, leading to quotas appearing to be duplicated. For example, the enigmatic leaf turtle, *Cyclemys enigmatica*, was accepted as split from the Asian leaf turtle, *Cyclemys dentata*, in 2017 at CoP17. In the publicly accessible records, all the quotas for *C. dentata* are duplicated for *C. enigmatica* for quotas set before 2017 (2014–2016), when the species was not even formally accepted. It would be wrong to assess *C. enigmatica* quota compliance against these quotas originally set for *C. dentata* as the species was not recognized at this time. This issue leads to instances where taxonomic changes occur and quotas are back-dated for new taxa before their accepted existence by the CoP. All quotas were manually checked for evidence of this issue and 518 quotas were removed where taxonomic splits led to duplicated historic quotas for newly accepted species (taxonomic updates that did not affect quotas of other species, for example, genus updates or synonym use were kept). This reduced the true number of quotas from 8,445 to 7,927.

We removed 97 quotas where the quota amount was given as '−1' or NA; these concerned specific unverifiable permit issues (*n* = 7,830). Similarly, we removed 69 unusable quotas, concerning quotas spanning

several calendar years (for example, March 2012 to March 2013) or concerned only re-exports, resulting in 7,761 taxa, exporter, years, term, purpose and source-specific quotas. To this we appended temporally accurate IUCN assessments for each species using the rredlist package[38] and data from a recently published assessment of whether species were likely to be threatened by international wildlife trade[39].

### Analysis

**Live quota compliance.** To assess quota compliance, we focused on quotas for live-traded species, which represent most quotas and are the least ambiguous term to compare with trade volumes. While trade records exist up to 2023, records for the most recent 2 years may be incomplete, hence we only consider 1997–2021. All instances of overlapping quotas for a given year and exporter were removed; this occurs where a species has a quota for wild-caught live specimens and another quota at another specificity, for example, live specimens from wild, captive and ranched sources. We summed Malaysian quotas where the three regions (Sabah, Sarawak and Peninsular Malaysia) were separately specified, as specific regions of export are not recorded in the CITES Trade Database. This resulted in a dataset of 3,158 quotas. We further removed all instances where species with live quotas also had quotas under other terms. For example, Asiatic softshell turtle (*Amyda cartilaginea*), Oriental rat snake (*Ptyas mucosus*) and Javan spitting cobra (*Naja sputatrix*) (from Indonesia) have low quota volumes for 'live' trade and higher volumes for 'skins and skin products' or 'consumption' yet much of the trade for skins is traded live but destined to be slaughtered for their skins[40] (rather than traded directly as skins as the quota note implies). This reduced the total number of usable live quotas to 2,712.

Traded volumes, as reported by the exporter, were accessed from https://trade.cites.org/ (ref. 41). Re-exports were removed and records filtered to include only the trade of live reptiles where the unit denotes the number of specimens (rather than weight). To each quota, we appended the traded volume for that exporter in that year, taking into account the specific combination of terms, sources and purposes specified. In addition to using exporter-reported trade volumes (presented in main text), we reassess compliance using importer-reported trade in full and report this in Supplementary Fig. 4. We also highlight records where exporter-reported volumes are quota compliant yet importer-reported volumes breach quotas (Table 1 and Supplementary Table 2). Such discrepancies could potentially be used to highlight further breaches or reporting errors and inconsistencies, although as a result of differing reporting practices between Parties exporter- and importer-reported the data can vary.

**Pre-quota to post-quota volumes.** To assess trade trends pre-quota to post-quota, live quotas were grouped into taxa, exporter, purpose and source quota time series. Quota series were filtered for taxon-exporter combinations where the same quota type had been applied for ≥5 years and where the Party and taxon concerned had been a signatory and listed for ≥5 years before the first quota, ensuring each quota had at least 5 years of pre-quota and post-quota data. The post-quota series was populated with data on the actual volumes traded and the quota levels specified. This was to enable contrasts between pre-quota volumes and set quota level and pre-quota to post-quota actual volumes contrasts. This resulted in 69 distinct species-exporter time series, covering 12 Parties and 68 species.

The year was zero-centred to the first year post-quota, for example, if quotas were implemented in 2005 then this became 0 and 2004 became −1. Traded volumes were zero-centred to the final year pre-quota, for example, 200 traded in 2004, becomes zero and if 100 were traded in 2005 this becomes −100. Owing to the large inter-taxon-exporter volume variances, we used group-level standardization rather than dataset-level. For example, a pre-quota value of −1 in 2001 denotes that the species was traded at 1 s.d. less than in

2004. Such standardization was needed to enable stable convergence even with the flexible hierarchical model used.

To this data, we fit a Bayesian hierarchal model with normally distributed errors, we allowed the scaled volume to vary by state relative to the pre-quota reference level volume ($\alpha$), where the post-quota trade state is shown by $\beta_1$ and the post-quota quota levels are $\beta_2$. This normally distributed and standardized model had superior posterior predictive checks compared to both Poisson and negative binomial models. These pre-quota and post-quota volumes were also allowed to vary through time ($\beta_{3-5}$). These fixed effect coefficients all varied per taxon (indexed by $j$) and the intercept varied distinctly by exporter (indexed by $k$) and the actual year of trade as a factor variable (rather than the zero-centred continuous year variable) (indexed by $l$) to account for temporal shocks or fluctuations.

$$
\begin{aligned}
\text{Volume} \quad &\sim N(\mu, \sigma^2) \\
\mu \quad &= \alpha_{j[i],k[i],l[i]} + \beta_{1j[i]}\,(\text{State}_{\text{Postquota-actual}}) + \beta_{2j[i]}\,(\text{State}_{\text{Postquota-quotas}}) \\
&\quad + \beta_{3j[i]}\,(\text{Year}) + \beta_{4j[i]}\,(\text{State}_{\text{Postquota-actual}} \times \text{Year}) \\
&\quad + \beta_{5j[i]}\,(\text{State}_{\text{Postquota-quotas}} \times \text{Year}) \\
\sigma^2 \quad &= \alpha
\end{aligned}
$$

$\beta_1$ can be interpreted as the counterfactual difference between business as usual (BAU) volumes ($\alpha$) and trade post-quota in year 0 (the year quotas were implemented). As we cannot know exactly what BAU volumes were going to be in year 0 this provides us with an estimate assuming the preceding trend continues. Thus answering whether quotas are temporally associated with absolute changes in volumes. A taxon-level $\beta_1$ coefficient of −1.5 could be interpreted as: quota implementation was associated with a 1.5 s.d. decrease in traded volume. Similarly, $\beta_2$ denotes the counterfactual difference between BAU volumes and quota limits. Likewise, $\beta_4$ is the change in temporal trend in post-quota volumes relative to pre-quota and $\beta_5$ is the same but for quota limits through time relative to pre-quota trends. All priors were zero-centred and diffuse.

**Frequency of quota updates.** To assess whether quotas represent effect adaptive management, we quantified how frequently quotas were updated. All quotas were grouped into taxa, exporter, purpose, term, source and unit quota time series. We removed any series that were composed partly or solely of zero quotas (bans). This resulted in 624 quotas time series at least 2 years in length (range 2–27, for example 1997–2023). We then calculated the number of times quotas changed relative to the preceding value. We hypothesize that quota compliance may be high because of the prevalence of rarely updated quotas, particularly among longer-term quotas. We parametrized a nonlinear Bayesian change-point model assuming normally distributed errors to explore if and at what quota length update stagnation occurred (where quota updating slows or stops despite increasing quota length) and estimated the corresponding pre-change and post-change association between quota length and the number of updates.

Priors for pre-change and post-change slopes assumed a normal distribution with a mean of 1 with s.d. of 2. In line with our a priori expectation that effective adaptive managed quotas are likely to have a 1:1 relationship between their length and the number of times, the quota has been updated. We set a broad normally distributed prior on the change point itself, reflecting our limited knowledge of its existence or timing, we specified a mean of 10 and s.d. of 5.

All models were run for 500 warmup iterations and 500 sampling iterations across four chains. Posterior predictive checks were used to assess fit and stable convergence (all scale convergence factors <1.02). We quantified model coefficients direction by calculating the direct probability of direction (pd) and term a clear directional association where 97.5% of the posterior shares the medians sign. Models were fitted with brms[42], posterior summarizing and testing using tidybayes[43]

and bayestestR[44] and all general data handling and plotting using the tidyverse ecosystem[45].

## Reporting summary

Further information on research design is available in the Nature Portfolio Reporting Summary linked to this article.

## Data availability

All data used in this study are publicly accessible and referenced in the main manuscript. The processed data used in the final analyses are available from https://github.com/OMorton/CITES_Quotas.

## Code availability

All analysis code is available from https://github.com/OMorton/CITES_Quotas.

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

## Acknowledgements

D.P.E. acknowledges funding from the Natural Environment Research Council (grant no. NE/R017441/1).

## Author contributions

O.M. led the conceptualization, data curation, methodology and formal analysis, wrote the original draft and edited subsequent drafts. V.N. contributed to the methodology and critically reviewed and edited subsequent drafts. D.P.E. contributed to the project conceptualization and supervision and critically reviewed and edited subsequent drafts.

## Competing interests

The authors declare no competing interests. However, V.N. is a member of the IUCN/SSC Asian Songbird Trade, primate, bear and mollusc specialist groups and was a previous member of the Dutch CITES Scientific Authority (2005–2017), giving him privileged information about the import and export of CITES-listed species into the EU. However, all data collected were derived from publicly available sources. This manuscript represents the views of the authors and does not necessarily reflect that of the CITES Scientific Authorities or IUCN/SSC groups.

## Additional information

**Correspondence and requests for materials** should be addressed to Oscar Morton.

# Reporting Summary

## Statistics

For all statistical analyses, confirm that the following items are present in the figure legend, table legend, main text, or Methods section.

| n/a | Confirmed | |
|-----|-----------|---|
| ☐ | ☒ | The exact sample size (*n*) for each experimental group/condition, given as a discrete number and unit of measurement |
| ☒ | ☐ | A statement on whether measurements were taken from distinct samples or whether the same sample was measured repeatedly |
| ☒ | ☐ | The statistical test(s) used AND whether they are one- or two-sided<br>*Only common tests should be described solely by name; describe more complex techniques in the Methods section.* |
| ☐ | ☒ | A description of all covariates tested |
| ☐ | ☒ | A description of any assumptions or corrections, such as tests of normality and adjustment for multiple comparisons |
| ☐ | ☒ | A full description of the statistical parameters including central tendency (e.g. means) or other basic estimates (e.g. regression coefficient) AND variation (e.g. standard deviation) or associated estimates of uncertainty (e.g. confidence intervals) |
| ☒ | ☐ | For null hypothesis testing, the test statistic (e.g. $F$, $t$, $r$) with confidence intervals, effect sizes, degrees of freedom and $P$ value noted<br>*Give P values as exact values whenever suitable.* |
| ☐ | ☒ | For Bayesian analysis, information on the choice of priors and Markov chain Monte Carlo settings |
| ☐ | ☒ | For hierarchical and complex designs, identification of the appropriate level for tests and full reporting of outcomes |
| ☒ | ☐ | Estimates of effect sizes (e.g. Cohen's *d*, Pearson's *r*), indicating how they were calculated |

*Our web collection on statistics for biologists contains articles on many of the points above.*

## Software and code

Policy information about availability of computer code

Data collection
> Data was directly downloaded from publicly accessible sites that are clearly linked to in the manuscript methods.

Data analysis
> All analysis was completed using R version 4.3.1
> Key packages used were:
> tidyverse 2.0.0
> rredlist 0.7.1
> brms 2.20.4
> tidybayes 3.0.6
> ggpubr 0.6.0
> viridis 0.6.4
>
> All code is publically available from github with the link clearly given in the manuscript.

For manuscripts utilizing custom algorithms or software that are central to the research but not yet described in published literature, software must be made available to editors and reviewers. We strongly encourage code deposition in a community repository (e.g. GitHub). See the Nature Portfolio guidelines for submitting code & software for further information.

# Data

Policy information about availability of data

All manuscripts must include a data availability statement. This statement should provide the following information, where applicable:
- Accession codes, unique identifiers, or web links for publicly available datasets
- A description of any restrictions on data availability
- For clinical datasets or third party data, please ensure that the statement adheres to our policy

> All data used in this study is publicly accessible and referenced in the main manuscript, the processed data used in the final analyses is available from https://github.com/OMorton/CITES_Quotas.

# Research involving human participants, their data, or biological material

Policy information about studies with human participants or human data. See also policy information about sex, gender (identity/presentation), and sexual orientation and race, ethnicity and racism.

| | |
|---|---|
| Reporting on sex and gender | NA |
| Reporting on race, ethnicity, or other socially relevant groupings | NA |
| Population characteristics | NA |
| Recruitment | NA |
| Ethics oversight | NA |

Note that full information on the approval of the study protocol must also be provided in the manuscript.

# Field-specific reporting

Please select the one below that is the best fit for your research. If you are not sure, read the appropriate sections before making your selection.

☐ Life sciences   ☐ Behavioural & social sciences   ☒ Ecological, evolutionary & environmental sciences

For a reference copy of the document with all sections, see nature.com/documents/nr-reporting-summary-flat.pdf

# Ecological, evolutionary & environmental sciences study design

All studies must disclose on these points even when the disclosure is negative.

| | |
|---|---|
| Study description | This study analyzed the diversity of taxa and trade types covered by current international trade quotas. We then further assessed national compliance with trade quotas using reported trade data. We analyzed national trade volumes pre- and post-quota implementation to assess how the setting of a quota affected trade volumes. |
| Research sample | Records of all specified quotas for internationally traded CITES-listed species, access from https://speciesplus.net/. This was augmented with the international trade data taken from https://trade.cites.org/. The combined data set represents all international trade in listed species and their compliance with internationally set quotas between 1997 and 2021. The data is representative of all legal international trade in CITES listed species to or from Parties, not of all trade. |
| Sampling strategy | No sample size calculations were used. All available quota data was used. |
| Data collection | Data was not collected by the authors data was collected by the UNEP-WCMC and the Parties themselves. OM downloaded and processed the data in R. |
| Timing and spatial scale | The data was "collected" from the online repositories in Autumn 2023. We used the 2023 version of the trade data, which will be the most recent and up to data until at least summer 2024. We used the maximum data range that can be appropriately used, starting in 1997 which was when the first international trade quotas were established and ending in 2021 which is the most recent year traded data should be used due to delayed reporting by some Parties. |
| Data exclusions | In the methods we clearly detail the limited circumstances where analyses focused on a subset of quotas. Namely these include the analysis limited to only live quotas (e.g. not skins, leather products etc.) other quota types were excluded as they cannot be objectively quantified. |
| Reproducibility | This was not an experiment. However we did do an additional robustness analysis using the importer reported data (main text used |

| | |
|---|---|
| Reproducibility | the exporter reported trade data). Due to differing reporter obligations these data sets are not expected to be identical, yet our results using either dataset lead to the same conclusions. |
| Randomization | NA |
| Blinding | NA |

Did the study involve field work? ☐ Yes ☒ No

# Reporting for specific materials, systems and methods

We require information from authors about some types of materials, experimental systems and methods used in many studies. Here, indicate whether each material, system or method listed is relevant to your study. If you are not sure if a list item applies to your research, read the appropriate section before selecting a response.

## Materials & experimental systems

| n/a | Involved in the study |
|---|---|
| ☒ ☐ | Antibodies |
| ☒ ☐ | Eukaryotic cell lines |
| ☒ ☐ | Palaeontology and archaeology |
| ☒ ☐ | Animals and other organisms |
| ☒ ☐ | Clinical data |
| ☒ ☐ | Dual use research of concern |
| ☒ ☐ | Plants |

## Methods

| n/a | Involved in the study |
|---|---|
| ☒ ☐ | ChIP-seq |
| ☒ ☐ | Flow cytometry |
| ☒ ☐ | MRI-based neuroimaging |

## Plants

| | |
|---|---|
| Seed stocks | NA |
| Novel plant genotypes | NA |
| Authentication | NA |

