## [Peer Review File · Nature Ecology & Evolution]

Peer Review Information

Journal: Nature Ecology & Evolution

Manuscript Title: Paper quotas: High compliance but ineffective quota-setting in the international wildlife trade

Corresponding author name(s): Oscar Morton

Editorial Notes:

Reviewer Comments & Decisions:

Decision Letter, initial version:

29th January 2024

Dear Dr Morton,

Your Article, "Paper quotas: High compliance but ineffective quota-setting in the international wildlife trade" has now been seen by three reviewers. I do apologise that it has taken us longer than we usually aim for to gather these reviews. You will see from their comments copied below that while they find your work of considerable potential interest, they have raised quite substantial concerns that must be addressed. In light of these comments, we cannot accept the manuscript for publication, but would be very interested in considering a revised version that addresses these serious concerns.

We hope you will find the reviewers' comments useful as you decide how to proceed. If you wish to submit a substantially revised manuscript, please bear in mind that we will be reluctant to approach the reviewers again in the absence of major revisions.

If you choose to revise your manuscript taking into account all reviewer and editor comments, please highlight all changes in the manuscript text file.

* Include a "Response to reviewers" document detailing, point-by-point, how you addressed each referee comment. If no action was taken to address a point, you must provide a compelling argument. This response will be sent back to the referees along with the revised manuscript.

2* If you have not done so already we suggest that you begin to revise your manuscript so that it conforms to our Article format instructions at <http://www.nature.com/natecolevol/info/final-submission>. Refer also to any guidelines provided in this letter.

[REDACTED]

If you wish to submit a suitably revised manuscript we would hope to receive it within 6 months. If you cannot send it within this time, please let us know. We will be happy to consider your revision so long as nothing similar has been accepted for publication at Nature Ecology & Evolution or published elsewhere.

Nature Ecology & Evolution is committed to improving transparency in authorship. As part of our efforts in this direction, we are now requesting that all authors identified as 'corresponding author' on published papers create and link their Open Researcher and Contributor Identifier (ORCID) with their account on the Manuscript Tracking System (MTS), prior to acceptance. This applies to primary research papers only. ORCID helps the scientific community achieve unambiguous attribution of all scholarly contributions. You can create and link your ORCID from the home page of the MTS by clicking on 'Modify my Springer Nature account'. For more information please visit please visit www.springernature.com/orcid.

Thank you for the opportunity to review your work.

[REDACTED]

Reviewers' comments:

Reviewer #1 (Remarks to the Author):

Overall comments

This has the potential to be an interesting paper but, in my opinion, much more work needs to be done for this to be the case. Quota coverage is not a particularly interesting research frontier; it is quick and simple to determine the CITES-listed species for which quotas have been applied using the CITES website. Compliance and adaptiveness have some merit as a research focus, but as presented the manuscript is superficial in its assessment of adaptiveness and the broader management of CITES-listed species for which quotas have been used. This is a major criticism of the paper and detracts from the paper being a potentially useful contribution to the literature. The manuscript would be much improved through greater effort to understand how quotas are set and managed; at the moment the authors appear to essentially guess how and why decisions are made on quota-setting, which is not a helpful basis for recommending improvements to policy and species and trade management. The recommendations also appear as an afterthought, are unclear, and do not appear to be fully articulated. This should have been addressed prior to submission.

Major points

1. The authors analyse trends in trade with respect to quota-setting but they fail to contend with the reality of trade and make big assumptions about the management of species for which quotas are applied. They deal with compliance adequately but then largely speculate as to the reasons why quotas have changed based on their analysis. There are many reasons why quotas may have changed or may not have changed beyond the 4 reasons the authors suggest in the section on quota compliance. The 4 reasons presented are also largely unsupported empirically and the manuscript does read like the authors are to a large extent guessing as to why the quotas may not have changed.
2. It is a shame that the authors did not liaise with key quota-setting countries (e.g., ID, MG, ET, GY and SR +/- others) to determine the rationale for quota-setting and how decisions related to quotas are made. This needn't have been for all species/quotas but could have focused on countries where non-compliance appeared to be an issue and/or those taxa for which quotas were exceeded the most. This would have provided useful insight to contextualise the other findings. There may or may not have been issues or new information which supported the application of quotas as they have been applied.

3Without such insight the authors appear to simply speculate about the reasons behind the trends their data analyses suggest but do not present convincing arguments. This is not robust and more work needs to be done on this front if this manuscript is to be published in a top journal.

3. At various points throughout the manuscript, the authors assume that quotas going unchanged over time is inherently negative for species, which may or may not be the case but little evidence is provided that genuinely or robustly supports this point. Deeper examination of how quotas are applied and the basis of decision-making through engagement with Parties could provide rich insight into quota management that would improve the manuscript.

4. The authors state that there is a lack of confidence in quota setting but provide weak support for this argument. The authors need to support this point much more substantially if they are to retain this in the article.

5. Regarding “paper quotas” the authors have not done enough to coin this term in my opinion. Many of the arguments used in the manuscript are based on speculation, selected examples, and are not well supported when it comes to understanding how quota-setting relates to the actual management of species by national CITES and other authorities.

6. Recommendations. The recommendations are unclear and read as an afterthought. It is not at all clear what the authors are suggesting in policy terms or who the audience is for such recommendations. These issues should have been addressed prior to submission.

Line by line comments

L16 - Who is “we”? Suggest the authors are clearer on who they are referring to here or edit this sentence and then the manuscript throughout. The authors, I’m guessing, are not referring to themselves.

L16-19 – the authors need to be clearer on how the 7000 quotas relates to 343 species. Have the authors treated an annual quota for one species for each year as one quota? This requires editing so it is clear to the reader.

L22-24 – does lack of update automatically infer “paper” quotas. I’m not convinced but I will see what the main text says on this.

L32-34 – though of course, this does not mean that these species are threatened by harvest/trade. Why use this 24% figure? Why not use the number of species actually in international trade? The World Wildlife Trade report estimated direct trade involved 12,000 species in a recent 5 year period. Suggest citing this report instead. Scheffers et al. conflate species being in trade with threatened by trade.

L34-35 – Conversely, if trade is well-managed it can bring myriad benefits to species and people. E.g., for rhinos, it can help support population growth rates. I suggest the authors make this point explicitly in the paragraph.

L37 – capital L needed?

L38-42 – Instead of saying “many”, why not include further detail on quota use. The CITES website has details of quotas for countries and species. I suggest the authors make this change, which will make it easier for the reader to grasp how widespread is the use of quotas.

L40 – mandated by who? Do you mean that quotas don’t appear in the Convention text or similar? Please clarify.

L43 – annual quotas can take the place of multiple NDFs; suggest making this explicit.

L47-48 – suggest revising syntax. This is an awkward sentence to read.

L51-52 – effectiveness would be more appropriate here rather than importance, surely?

L54 – complied with by whom? Trade chain actors as well as agencies implementing CITES? Suggest making this clear and explicit.

L57-58 – what do you mean by rigidity here? Parties can set some quotas unilaterally so in theory could be used for many species at scale. Please elaborate on rigidity so it is clear for the reader.

L66 – do you mean thousands of individuals? Please clarify.

L68 – I’m not convinced a key research frontier is understanding quota coverage. The species-country combinations that use quotas can be easily downloaded from the CITES website. Compliance and adaptiveness have some merit as research avenues but see other points on how this has been done.

L89-89 – this should say “Res. Conf. 14.7...”. Please correct.

L105 – I would amend the wording here. Zero-quotas, while having the effect of a ban, are not the same as a ban in CITES terms. They may be changed much more easily and the burden of proof to do so is much lower than would be needed to transfer a species from CITES Appendix I to II for instance.

Fig 2E and throughout – As above, I would refrain from referring to zero-quotas as bans. Proxy bans may be appropriate.

L104-110 – So, compliance is high – this is a good thing.

L117-118 and Fig. 2G – This is also a good thing; most quotas are not being exceeded.

Table 1 caption – check syntax.

Fig. 3a – It is not clear why the y-axis scale is used. Please explain/justify for the reader.

L213-218 – This paragraph focuses on the negative aspects of quota use when one of the key results of this manuscript is that there are high levels of compliance with quotas. I strongly suggest the authors re-frame the paragraph to explicitly recognise the high levels of compliance.

L214 – what does vague coverage mean? Please elaborate.

L214 – inappropriately high volumes – while this may be the case it is for a small proportion of species only. This should also be stated explicitly.

L221-231 – this paragraph epitomises one of the main criticisms I have of this paper. That is, while the authors analyse trends in trade with respect to quota-setting they fail to contend with the reality of trade and make assumptions about the management of species for which quotas are applied. For instance, why did the authors not liaise with key quota-setting countries (ID, MG, ET, GY and SR) to determine the rationale for quota-setting and how decisions related to quotas were made? This needn't have been for all species/quotas but could have focused on countries where non-compliance appeared to be an issue and/or those taxa for which quotas were exceeded the most. This would have provided useful insight to contextualise the other findings. There may or may not have been issues or new information which supported the application of quotas as they have been applied. At the moment this paragraph reads like the authors have looked at the data and just made assumptions about management of quotas, including for the two species examples used, whereas liaison with exporting countries could have provided insight making the dataset richer and the article much more compelling.

L222 – is this monitoring of trade or species in the wild, or both. Please state explicitly so it is easy for the reader to grasp.

L243-252 – I don't doubt that this is an issue but it doesn't affect many species (35?), which is a very small proportion of CITES-listed species. Again, deeper exploration of the management of such species – through liaison with relevant agencies and stakeholders - could reveal interesting insight into what is happening. This includes where such trade takes place because of genuine corruption or because of other factors e.g., lack of capacity with the relevant CITES authorities – both importers and exporters.

L255-256 – This is not inherently negative for biodiversity where quotas are well evidence-based but the manuscript provides little detailed insight into the extent or nature of management for species subject to quotas.

L255-261 – Again, this is where some investigation into the management of species with quotas could have helped inform the discussion. There are likely many other reasons why quotas may not have changed beyond the 4 reasons the authors suggest. These 4 reasons are also largely unsupported empirically and the manuscript does read like the authors are to a large extent guessing as to why the quotas may not have changed.

L268-270 – See above comments on digging deeper into the management of species for which quotas are set. Such justification may or may not exist for different species and country combinations but we don't know and the authors don't present much evidence in support of their arguments here.

L277-281 – The authors fail to contend with the reality that many CITES authorities are chronically under resourced, which could explain some of the issues with quota setting and a lack of “hard” data and evidence where there are problems with levels at which quotas are set.

L277-281 – the main point made here is not well supported by literature. The authors include 3 references only, two of which are dated and for which circumstances may have changed in the last 20 years. The authors need to support this point much more substantially if they are to retain this in the

article.

L281-284 – The authors have not done enough to coin this term in my opinion. Many of the arguments used in the manuscript are based on speculation, selected examples, and are not well supported when it comes to understanding how quota-setting relates to the actual management of species by national CITES and other authorities.

L285 – Again, insight into the management of species could reveal interesting insights. Have quotas been set because species are about to be traded and/or countries are preparing for trade in particular species from wild and/or captive sources?

L285-286 – A counter-argument could be that Parties have more up to date information than is currently available publicly and have therefore justified higher quotas, but this isn't captured by the authors because they haven't engaged with the relevant authorities.

L289-291 – It is a shame the authors didn't consider engaging some CITES Parties when conceiving and carrying out this study, because doing so would have enabled them to answer this question, at least in part.

L294 – What does further discussion mean? By whom, with whom, with what purpose, and in what context? Does this relate to CITES or is this discussion nationally? Or otherwise?

L295-298 – What evidence? Please be explicit. Acceptable to who? Please be explicit. What is the rationale for publicly available evidence? And, how realistic do the authors consider this to be, both within and outside the CITES context.

L296-297 – As per a previous comment, lack of an update to a quota is not necessarily cause for conservation concern. The authors need to address this point here and throughout the manuscript. This includes at L304-305; dialogue with key Parties could have been insightful here.

L298-301 – Systematic policy change? Which policies and what change are the authors calling for? This reads as an afterthought and it is not at all clear what the authors are suggesting in policy terms.

L302-309 – So, what changes are actually being proposed? It is not at all clear. For example, are the authors proposing a new CITES Resolution to guide Parties, revision of existing guidance on the use of quotas or something entirely different? The authors need to elaborate on their recommendations so the reader can grasp what is being proposed.

L306 – Amendments to what? Importing countries already report trade in CITES-listed species, at least for some species.

L308-309 – In my opinion, the authors cannot claim that this is categorically the case regarding quotas because they failed to ask relevant CITES agencies about their management practices and otherwise only speculate on the reasons for trends in trade of species with established quotas.

Reviewer #2 (Remarks to the Author):

This is a well-written manuscript identifying important issues around the setting and managing of quotas for the international wildlife trade. The study uses empirical evidence to highlight questionable patterns in quota setting and management that deserve more attention and scrutiny. I believe this manuscript merits publication and that a wide audience would be interested in its findings. I have made suggestions mostly for improving clarity and readability of the findings.

-The term “we” is used a few times, consider describing the community / people being referred to here.

-Global comment: The percentages provided vary throughout the manuscript in terms of how many decimals are used (some are whole numbers, some to the tenth, others to the hundredth); personally I think rounding to whole numbers would be appropriate for this work (in text and in tables/figures).

-Lines 15-16; Perhaps omit “sustainable” here in the description of a quota and instead list “sustainability” as an area where we lack sufficient global assessments.

-The “quota coverage” section answers the “what” quotas cover but does not seem to answer the “where” as specified in line 73.

Line 78-80 - Somewhere in the beginning of the “quota coverage” section, I suggest having one sentence that defines the key terms you will be referring to without along with examples, specifically for: “term”, “source”, and “purpose” (you list examples for “term” and “source”, but not for “purpose”, I would put all together in an explanatory sentence; i.e. not one with any findings).

-In the “quota coverage” section you say “term-monitored” (line 80) and on line 86 you say “type of item managed” - I think it would help the reader if you were consistent in your terminology.

-Similarly, in the text you refer to the “species-source” (line 80-81) and in the figure it only refers to “source” - would be helpful if consistent (maybe just delete species).

Line 88-90: Perhaps provide a bit more explanation here rather than just saying “and presumably referred to to wild trade” with citation.

-Lines 96-97: In the figure I would provide a quick explanation of the terms used (source, purpose, term).

Fig. 1A - Decimal places vary for bar values, I suggest whole numbers or one decimal place. Also, consider consistency in how the bars are labeled, i.e. perhaps “source” goes first in each bar label, then “purpose”, etc. so the reader can more quickly absorb what we are looking at. Not sure why the bars are in the order they are in? Could be helpful to order from highest to lowest or vice versa.

Fig. 1B - I think the y-axis should be “Term” to be consistent.

Figure 1 - I am not sure why some of the bars are red? Should mention in the caption. Also, I know the definitions of the various abbreviations are in the supplementary but I think it would help the reader if the terms (or the most commonly used ones if space is an issue?) were also defined in the caption or included in the figures - it is hard to digest with all the unknown acronyms. I would also have the same x-axis for 1b/c/d to enable easy comparisons for the reader.

Line 115 - Perhaps qualify quota breaches as “Recorded” or “Known” quota breaches given there could likely be breaches for which there is no documentation

Line 118-120 - For this example perhaps provide the number of individuals as well, not just % to give a sense of magnitude.

Lines 135-136 - The inset panel does not make visible the full distribution (i.e., there is nothing visible after 150% or so...)

Table 1 - Suggest spelling out “Exporter-reported” and “Importer-reported” in the top row for ease of reading / understanding figure.

Line 145 - Says “Using 69 species-party quota time series” - this is a bit unclear, perhaps use plain language to say national-level quotas.

Lines 152-153 - I would clarify so it is clear you are saying that 46% of quotas were set “above” the pre-quota trade volumes.

Lines 153-155 - Not a complete sentence, need to better explain this point.

Lines 167-169 - Seems an indirect way of saying that zero quotas (or bans) worked for this species?

Line 174 - should be “volume” not “volumes”

Lines 176-177 - Caption for “B” is listed twice, combine and make order consistent with caption for “C” (i.e. describe the summary and then the conceptual figures).

Figure 3 - The conceptual figures of how quotas and traded volumes changed after quotas are great (in 3B/3C); I think it would be helpful to label them with letters (e.g. A through R) and then refer to these letters when discussing a specific pattern in the text (e.g., on line 159). I would also consider adding % at the top of each of the bars and possibly even the volumes within each bar.

Line 214-215 - Instead of “vague coverage” perhaps “imprecise data”; I would add “quota” so reads “inappropriately high quota volumes”; and also add quota to “lack of updating quotas”

Line 224-225 - Delete “is” so reads “rendering compliance impossible”

Line 226 - Should it just be “10,000” - no s? Or instead “10,000-plus”?

Line 227 - Suggest adding “per year” after live individuals

Lines 232-234 - Suggest making first sentence more straightforward, less wordy

Line 255-256 - perhaps add “set” so reads “most commonly set higher”; instead of “go” should be past tense

Line 261 - You are saying that in most cases it seems the original quota was based on inaccurate data? I would say this there to clarify.

Line 267 - Suggest deleting “exploited”

Line 277 - You say there is a clear “lack of confidence” is it confidence or instead a “lack of evidence”

289-291 - Nice

298-301 - suggest rewording so sentence starts with “Systemic policy change is necessary to produce evidence to the contrary.” I am not sure you need the next fragment about an avoidable situation? Could just go right into the changes that must be addressed starting on 302.

308-309 - great closing sentence

Supplementary line 29: typo of “otherwisree”

Reviewer #3 (Remarks to the Author):

The key theme the authors put forward around “Paper Quotas” is an important topic - and the paper makes an important point around the need to scrutinise the export quotas for the international trade in wildlife more systematically in the CITES context to ensure they are fit for purpose. To my knowledge, this is novel and not something that has been looked at systematically in this way before. I do see this topic to be of interest / relevance to those that work on CITES and that are making decisions in the context of international wildlife trade. With overexploitation one of the two key drivers of biodiversity loss, the is a key topic to explore.

I do wonder, however, if the authors are missing some key aspects of this argument - for instance, they do not look closely at which countries are issuing quotas -- and which are not. Are there populations of globally threatened species that occur in Range States that aren't issuing quotas at all? Or are there countries that are only issuing quotas for only a small number of species e.g. <XX over the last ten years...? The authors should also highlight up front that quotas (in most cases) are voluntary – hence there will be many countries that are not issuing any quotas which may be an even bigger conservation concern than whether the ones that are published are not set at the right level. This “country lens” seems like a key aspect to include – or at least to say that future analyses / focus should look into this aspect in more detail to add further insights into whether CITES quotas are fit for purpose.

A key part of the argument is that the quotas are too high -- the evidence around how the authors know they are too high could be strengthened. Just because trade is lower than the quota after it was put in place, doesn't mean the quota is too high -as there are CITES Authorities in each country that are issuing CITES permits and so can control the trade levels and ensure it stays below quota. It also could be that the demand is just not there –so the trade is lower and never reaches the quota level. I’m not saying this is the case – as there is certainly lots of trade that is unsustainable – but providing more evidence

11around the sustainability aspect is important to prove that they are “paper quotas”. The paper would benefit from a more detailed look at why the quotas are believed to be unsustainable. Are there specific examples where it is clear that the level of trade is both below the quota and that the trade levels are unsustainable (i.e. the quota is set too high and leading to population declines driven by trade). I’m sure there are many – but this does not come across clearly in the paper. Think this argument – which is key - needs more examples. Bringing in the IUCN Red List status and the population trend could help make this case (e.g. some further exploration of globally threatened species with stable or declining populations where quotas seem inadequate to promote sustainability).

Overall, I think this is an important topic and a novel analysis, but I would suggest that it could be strengthened in places to make it more evidence-based and compelling around the need for an overhaul of / more scrutiny on the CITES quota system.

Detailed comments on specific lines / sections are provided below:

Abstract:

Lines 20-22: Suggest checking the 62.3% statistic as this seems high – in the section where this is referred to in the text, it mentions Brookesia – and I just wonder if some of this increase relates to quotas for species that may have had a nomenclature change (and so wouldn’t have had reported trade previously). Many of the chameleons have split or lumped over the years. This may skew the results?

23 - “were never updated” – this wording is inaccurate in the CITES context-- as technically CITES export quotas are communicated by the CITES Party (government) to the CITES Secretariat and “published” anew on the CITES website (and Species+) each year. It would be more accurate to say “have remained at the same level each year”.

Introduction:

31-32 – Suggest adding an explicit mention around conserving the species themselves; this seems to be a gap, but is a key provision of CITES (e.g. ensuring international trade is not detrimental to the species populations in the wild), which you say later.

37 – “Listed” isn’t typically capitalised (normally written as “CITES-listed” or “listed in the CITES Appendices”. Could expand to be more clear – e.g. “regulates international trade in the over 40,900 species listed in the CITES Appendices.” Source for total number - <https://cites.org/eng/disc/species.php>

45-46 - As an example -- other reasons may apply e.g. as part of Review of Significant Trade process.

70 – birds - - suggest being a bit careful here - there is still quite a large number of birds traded - internationally and domestically - and large groups (e.g. songbirds) that aren't listed and so there is uncertainty about the scale of this trade. See <https://cites.org/eng/news/technical-workshop-songbird-trade-conservation-management-2023> -- e.g. millions of songbirds in trade...

71 – “all other vertebrate trade combined” – needs caveating to say “greater than for all other CITES-listed vertebrate trade combined” (otherwise, the bird issue arises – and perhaps other vertebrates – as per comment above).

79 - Lots removed from original 8470 Species+ download -- when species were removed due to duplication / nomenclature issues - just want to ensure that species that should still be there – e.g. if it has a note that it was originally published for the synonym those have still been retained (e.g. *Chelonoidis carbonarius* and *Rhampholeon spinosus*). (understand that a lot had to go, but was larger than I expected).

79-81 -- As all readers may not be as familiar with quotas, I think it would also be important to state that the species (or taxon) and country and year were also specified in each of these

83-84 – Suggest adding summary stat on countries. As countries are so essential to quota setting, it would also be worth adding to total number of countries issuing export quotas - I get 70 (I ignored "Sudan [prior to succession of South Sudan]" to avoid double counting). To me this another important angle -- are there any key reptile range states that are trading and don't issue quotas at all -- or that haven't in the last 5-10 years? Is this a cause for concern? What about the discrepancy between countries -- you'll note that some of the countries that have had sustainability issues flagged in the context of the Review of Significant Trade process have large numbers of quotas -- are there others with fewer published that are actively trading Endangered / Critically Endangered species that don't regularly issue quotas (e.g. have <20)? Note also if you sort by countries, the top five countries represent more than half of the reptile quotas --seems notable... (links to the Review of Significant Trade again...).

90 – I think the purpose of the trade is not specified in 93.8% cases because the purpose of the trade doesn't really matter in practice in this context – the assumption is that it's for all purposes (e.g. a live animal traded for commercial purposes or a zoo is still a live animal that has been taken from the wild (assuming source is wild), so the trade would count towards the quota). I also suspect that the purpose is not requested by CITES – so the 6.2% that provide a purpose are likely to be very specific purposes...

Main point being, I wouldn't make too much of this finding as not sure it has a huge impact on conservation / wildlife management...suggest removing the "purpose" aspect of the analysis or at least downplaying it as I do not see this as crucial to making quotas effective in CITES.

Figure 1 – as per above – suggest removing "Purpose" considerations. Also – year range says to 2021, but elsewhere you say 2023.

Table 1 – suggest putting "potential breaches" as this trade was only over quota on the importer side (and there are reasons why there could be discrepancies). Also – consider putting the IUCN Red List category in this table – were any of these EN / CR species? Add key for what "ER" and "IR" are – or add "ER" and "IR" into table header.

122- species with quotas for which there wasn't any trade? (rather than quotas with no trade...)

155-157 – Not sure about the sentence about Brookesia – was the low trade pre-quota due to nomenclature changes with the genus – or even the Review of Significant Trade Process? There has been a lot of taxonomic revision in this family. Perhaps worth a check before inferring that it may be a success story.

Figure 3 – I found this figure a bit challenging to interpret. To me, section "A" (top graph) seems quite good from a conservation perspective if I'm interpreting it properly -- looks like the trade trend is upward "pre-quota", but then subsequent to the quota setting, the trade stays largely lower than the quota and starts to decrease? Though this is contrary to the overall message of the paper... Also the Figure header has two "B"s and two "C"s" – which I understand is likely top figure and then the part of the figure below - consider revising to ensure it is clear.

186 – 190 - Each quota is communicated to the Secretariat / published anew each year (even if it maintains at the same level). The way this is currently characterised seems a bit misleading...

191-192 - How many of these are COP approved quotas -- including zero quotas - which would make sense for them to stagnate as they are agreed via the CoP...

Discussion

213- As there are different types of quotas – e.g. harvest quotas, hunting quotas, and export quotas, etc... - suggest tightening up the language so it's clear they are "export quotas" throughout. For instance the first sentence of the Discussion would be more clear if you say "international export quotas

published for CITES-listed reptiles.”

214 – one of the main points is around “, inappropriately high volumes,” – but in the section that follows (“Quotas as a tool for sustainable use”) – I don’t think this argument is effectively made. Which species are at risk from high quotas? Which species have trade levels that are too high and leading to population declines? Which ones are globally threatened, with trade as an on-going threat where the quota is set too high. These seem like important angles for this argument, but not sure the Discussion adequately covers this.

218 – Similarly, tightening up the language around “international trade” (rather than just trade) is also important throughout – e.g. the last sentence should be “to ensure international trade is sustainable in ...”. I’d suggest reframing that way (rather than “sustainable use”) as in many cases "ensuring sustainable use" is not the goal (e.g. for species where zero quotas are needed), but rather ensuring that international trade is sustainable. Check throughout (e.g. header of next section).

223-226 - Are there other, more current examples? This Nicaraguan spectacled caiman example is a bit old at this point as the last time a Nicaragua issued a quota for this was back in 2010 -- since then, it has been "skins, wild taken" - which is pretty clear -- so things have improved!

242- Perhaps expand further beyond just “then quotas just...”-- e.g. quotas without a firm scientific basis...compound the issue... or "inappropriately set quotas" (e.g. those that are too high?... Think it's worth being explicit to get your point across.

249 – as there is some uncertainty – suggest reframing “breached” to “apparently exceeded” (see next comment)

251 - That "potentially" breach (as well as being “potentially illegal”) -- suggest being a bit careful here as there are reasons why discrepancies may occur in the CITES trade data (e.g. year-end trade, importers reporting on the basis of "permits issued" and trade may have been cancelled), etc... so it may not be a breach. See page 14 of the CITES Trade Database Guidance on potential reasons for discrepancies between importer/exporter parties -- and add to the References if you include (citation is specified in the doc). https://trade.cites.org/cites_trade_guidelines/en-CITES_Trade_Database_Guide.pdf.

255-267 – In general it’s a good thing that quotas are higher than current trade volumes – that could be interpreted that CITES is working – countries are implementing CITES well as they are keeping their trade under the quote. In terms of the scenarios – I think there needs to be some recognition that one

scenario is that CITES Parties are keeping trade levels at or below their published quotas rather than demand necessarily declining. In other words, it's important to consider that CITES Authority will stop issuing permits for trade when they reach their quota (ideally!). For scenario 2) not sure I fully understand why other threats would make the quota irrelevant? Explain this more -- are you saying other threats have made the species population decline and that's why they're not being traded? Not sure... For scenario 4 – many species will lack data, so if you're expecting perfect data for the thousands of species, this is unrealistic – I don't think that means the quota is irrelevant (this is where adaptive management would come in...?). Important to strengthen your argument around the sustainability concerns if you want to say this.

285 – A bit worried about criteria 1. Sometimes there are cases where it may be precautionary to have a quota even in the absence of trade. For example – a zero quota. Are any of the 59 zero quotas?

Conclusions

295 – suggest addition of another paper beyond the lead author on this paper

295-296 - Strangely phrased (“it cannot be acceptable...”). Would be stronger if the paper had put forward compelling examples of where the legal trade is unsustainable – and there are many! Then the Conclusion could more clearly draw out the flaws in the current system, etc...

299-301 – I don't think this sentence makes sense “The onus must be on systematic policy change to produce evidence to the contrary – a completely avoidable situation if proactively evidencing sustainability had truly been the status quo rather than reactively highlighting unsustainability.” [Also be careful as this paper could be characterised as "reactively highlighting unsustainability"]

302-304 - Not sure the “vagueness” of the quotas is really an issue in practice (particularly on “purpose” as highlighted above) – and I do think that quotas are getting better defined in recent years (see caiman example I mentioned above) -- not sure the vagueness is really the heart of the problem here. That said, if authors disagree, they could engage directly with the CITES process and suggest changes? Seems like the main issue is ensuring that the quota is set at a scientifically justifiable level that means that trade is not having a detrimental effect on the species in the wild – and where there are declines, the quota should be revisited to determine if a lower quota is needed. This to me, is the most important and compelling argument to focus on. Suggest considering strengthening this conclusion more (currently missing).

306-307 – importers do already report on Appendix I-listed species. Might just want to specify “all CITES-listed trade”.

308 – not sure this makes sense “to add onus to importer responsibility, rather than just exporters.”

309 – Same point as above on last sentence –(Technically, sustainable trade CAN exist without evidence...). Ending could be strengthened. Something that really drive home that as the world trades more and more species – and with biodiversity in sharp decline – a stronger and more evidence-based framework for international quota setting and compliance is needed in order to really ensure international trade is sustainable...

367-368 - Fine to remove these and call them duplicates for your purposes, but it's incorrect to call it a mistake - Species+ is set up to show the history and inform CITES implementation, so these are presenting the facts around the quota issuance and provides CITES Parties with important context around CITES nomenclature and historic quotas, etc. (e.g. that a species would have been subject to the quota prior to the split / lump / taxonomic change). Please revise “mistakenly back-dated for new taxa prior to their accepted existence by the CoP.” Technically, the species did exist, they were just called something else (and traded under that other name).

376-379 - why don't you use the RL status or Appendix listing in the analysis? (sorry if I missed it?)

386 – 2021 vs. 2023 – think you say 2023 in the manuscript in places – needs check.

390 – Please refer to this as the CITES Trade Database (not the “CITES database”)

391-397 – Not sure this is clear why they were removed – quotas can be published under different terms and you could still compare the trade in live with the quota for live...

398 - Please add in the name of the database – the CITES Trade Database – along with the link. There are instructions on how to cite in the CITES Trade Database Guidance.

442 - Unclear why these are removed -- won't this miss out on lots of species with sustainability issues - e.g. those that have gone through the Review of Significant Trade Process and then have CoP-approved quotas?

444-446 – As in section 255-267, think you need more evidence around them being “irrelevant” ---

Compliance could also be high because these are voluntary quotas issued by the countries and there are CITES Authorities managing national trade levels, issuing permits and ensuring trade stays within their national quotas...

Author Rebuttal to Initial comments

Response to reviewers

Reviewer #1

Overall comments

This has the potential to be an interesting paper but, in my opinion, much more work needs to be done for this to be the case. Quota coverage is not a particularly interesting research frontier; it is quick and simple to determine the CITES-listed species for which quotas have been applied using the CITES website. Compliance and adaptiveness have some merit as a research focus, but as presented the manuscript is superficial in its assessment of adaptiveness and the broader management of CITES-listed species for which quotas have been used. This is a major criticism of the paper and detracts from the paper being a potentially useful contribution to the literature. The manuscript would be much improved through greater effort to understand how quotas are set and managed; at the moment the authors appear to essentially guess how and why decisions are made on quota-setting, which is not a helpful basis for recommending improvements to policy and species and trade management. The recommendations also appear as an afterthought, are unclear, and do not appear to be fully articulated. This should have been addressed prior to submission.

Thank you for the comments, we believe our additional analyses have addressed these concerns.

Major points

1. The authors analyse trends in trade with respect to quota-setting but they fail to contend with the reality of trade and make big assumptions about the management of species for which quotas are applied. They deal with compliance adequately but then largely speculate as to the reasons why quotas have changed based on their analysis. There are many reasons why quotas may have changed or may not have changed beyond the 4 reasons the authors suggest in the section on quota compliance. The 4 reasons presented are also largely unsupported empirically and the manuscript does read like the authors are to a large extent guessing as to why the quotas may not have changed.

We have removed the four points referred to here and rewritten large sections of text completely.

*“Similarly, quotas that remain unchanged through time could be sustainable but given rapid and severe environmental change (via deforestation, illegal hunting, climate change, etc), Parties must justify the relevance of historic quota levels with up-to-date data, especially for species threatened by multiple stressors (Symes et al., 2018). This specific issue is identified in CITES Resolution Conf. 14.7 (Rev. CoP15) where the document highlights the concern that quotas may not be updated to reflect changing climatic threats such as droughts (CITES, 2013b). Alternative options include replacing fixed quotas with adaptive quotas directly linked to monitoring of species populations like South Africa’s recent shift from fixed trophy quotas to 0.5% of the population for black rhinoceros (*Diceros bicornis*) exports (CITES, 2019), maintaining socioeconomic benefits and population integrity (‘t Sas-Rolfes et al., 2022).”*

2. It is a shame that the authors did not liaise with key quota-setting countries (e.g., ID, MG, ET, GY and SR +/- others) to determine the rationale for quota-setting and how decisions related to quotas are made. This needn’t have been for all species/quotas but could have focused on countries where non-compliance appeared to be an issue and/or those taxa for which quotas were exceeded the most. This would have provided useful insight to contextualise the other findings. There may or may not have been issues or new information which supported the application of quotas as they have been applied. Without such insight the authors appear to simply speculate about the reasons behind the trends their data analyses suggest but do not present convincing arguments. This is not robust and more work needs to be done on this front if this manuscript is to be published in a top journal.

This point is raised in multiple specific comments by the Reviewer so we address all the concerns here, rather than repeating ourselves throughout the response.

Primarily, our disagreement with this approach stems from it being unfeasible to do at scale and through time. Hundreds of quotas are set annually, concerning tens to hundreds of species and Parties. VN on the author team has worked with national CITES authorities and previously reached out to them regarding analyses –exactly as suggested here. At best, he received an acknowledgement of receipt but no further response. Equally important, the process of contacting a limited number of Parties for a set number of species generates bias regarding who decides to respond (if anyone), the qualitative nature of their responses, and potentially their impartiality, in addition to the failure to capture the temporal component of changes in quotas which we seek to understand.

Nevertheless, if the Editor insists, then we are willing to try to contact a number of MAs and SAs from a selected number of countries to provide general feedback on specific quotas.

3. At various points throughout the manuscript, the authors assume that quotas going unchanged over time is inherently negative for species, which may or may not be the case but little evidence is provided that genuinely or robustly supports this point. Deeper examination of how quotas are applied and the

19basis of decision-making through engagement with Parties could provide rich insight into quota management that would improve the manuscript.

We deal with this comment in the specific points raised by the Reviewer below. In the changing world where threats and populations vary through time it is the expectation that the number of individuals that could be sustainably removed would also vary annually. National UK fishing quotas for example change annually (<https://www.gov.uk/government/statistical-data-sets/fishing-quota-allocations-for-england-and-the-uk>) never remaining identical. But we specifically acknowledge that quotas remaining unchanged is not inherently negative, but may indicate a lack of adaptive management. We also add further text on this issue from CITES Resolution Conf. 14.7 (Rev.CoP15) where the Secretariat highlighted this very issue that quotas may not be being frequently updated to respond to changing conditions. Thus this is very clearly a potential issue.

“Parties must justify the relevance of historic quota levels with up-to-date data, especially for species threatened by multiple stressors (Symes et al., 2018). This specific issue is identified in CITES Resolution Conf. 14.7 (Rev. CoP15) where the document highlights the concern that quotas may not be updated to reflect changing climatic threats such as droughts (CITES, 2013b).”

As stated above, the author's (VN) prior experience engaging with Parties relating to previous research articles has been unsuccessful. Furthermore, how to treat any information collected and its inherent biases poses a challenge. But, as above, if the Editor would prefer us to do so, then we are willing to contact a limited number of SA's and MA's.

4. The authors state that there is a lack of confidence in quota setting but provide weak support for this argument. The authors need to support this point much more substantially if they are to retain this in the article.

This is a strongly supported claim, both that there is limited confidence in NDFs and in quotas. We have added more references than the considerable number we already had to support our argument. If the Reviewer can point us to specific references quantifying/highlighting broad-scale confidence in quotas/NDFs, we would be happy to engage with them.

“Yet, there seems little evidence that NDF practices effectively assess offtake sustainability in the first place (CITES, 2020; Morton et al., 2022). A review of accessible NDFs (totalling 36) by the CITES Secretariat highlighted only 41.7% fully considered population trends, 44.4% incorporated other threats, and 36.1% fully considered how the precautionary principle would apply (CITES, 2020). For instance, Thailand made positive NDFs for several seahorse species, but subsequently failed to support these with evidence (Foster & Vincent, 2021). Independent assessments for two species concluded a negative NDF for one, and changes to current practices for the other (Aylesworth et al., 2020; Foster & Vincent, 2021). If NDFs are not accurately assessing sustainable offtakes, then inappropriately set quotas just compound this issue.”

“Across many taxa and over time, there is a clear concern that quotas are not always based on robust data or set sustainably (Dumenu, 2019. Hughes et al. 2023). Instead, some authors suggest that quotas can be based entirely on subjective information (concerning Aquilaria quotas; Newton & Soehartono, 2001), guesswork rather than hard data (concerning Indonesia’s quotas; Soehartono & Mardiatuti, 2002), or at odds with the principles of sustainable use (for leopard quotas; Trouwborst et al., 2020).”

5. Regarding “paper quotas” the authors have not done enough to coin this term in my opinion. Many of the arguments used in the manuscript are based on speculation, selected examples, and are not well supported when it comes to understanding how quota-setting relates to the actual management of species by national CITES and other authorities.

We have removed this term, including from the title which now reads *“High compliance and coverage but insufficient adaptive setting in international wildlife trade quotas”*.

Over the course of the Reviewers specific comments, we have updated much of the text and are now more specific regarding what could be improved.

We strongly disagree with the Reviewer's assertion of *“selected examples”*. All of our statements regarding concerns or issues with NDFs, quotas and other CITES processes are specifically supported by multiple references. If the Reviewer can suggest any sources we have not engaged with we are happy to add them.

6. Recommendations. The recommendations are unclear and read as an afterthought. It is not at all clear what the authors are suggesting in policy terms or who the audience is for such recommendations. These issues should have been addressed prior to submission.

We have rewritten this and now make very explicit recommendations to revise CITES Guidelines and suggest a specific but larger-scale potential amendment. While acknowledging the difficulty of this suggestion.

“We echo multiple prior calls for serious international discussion and potential reforms to CITES to more robustly embed an evidence-based framework and justified quotas in national practice (Hughes et al., 2023b, 2023a; Morton et al., 2022).”

“Systematic policy changes up to and including amendments to the Convention are needed to proactively support the evidenced sustainability of trade. We suggest the following amendments would improve assessing compliance and confidence in sustainable offtake. Firstly, establishing and enforcing a minimum acceptable level of information (e.g. 1 or more comparable terms) as potentially ambiguous quotas only further entrench uncertainty, this would likely require a minor revision to the Annex

Guidelines of Resolution Conf. 14.7 (Rev CoP15). Secondly, requiring Parties to submit NDF's supporting quotas each time they are communicated to the CITES Secretariat, thus allowing scrutiny of the quota's validity and relevance to current populations and threats. The Global Biodiversity Framework 2030 Target 5 focuses on the sustainable use and trade of wild species yet has no listed data indicators to capture the legal wildlife trade (excluding fish stocks), creating such data must be a priority. Such an amendment would be significant, and even if accepted by the majority of Parties, slow to enter into force, but by identifying capacity and implementation gaps in Parties NDF practices, such NDF reporting could be used to leverage increased funding for key Parties from global sources (e.g. the Global Biodiversity Framework Fund)."

Line by line comments

L16 - Who is "we"? Suggest the authors are clearer on who they are referring to here or edit this sentence and then the manuscript throughout. The authors, I'm guessing, are not referring to themselves.

Corrected.

L16-19 – the authors need to be clearer on how the 7000 quotas relates to 343 species. Have the authors treated an annual quota for one species for each year as one quota? This requires editing so it is clear to the reader.

Edited for clarity. A quota refers to a specific quota as submitted to the secretariat referring to a number of individuals (or other term) of a species (or genus/subspecies) to be exported from a Party in a given year (sometimes from a specific source or for a specific purpose).

"Using over 7000 country-year specific reptile quotas established under the Convention on International Trade in Endangered Species of Wild Fauna and Flora (CITES) covering 343 species"

L22-24 – does lack of update automatically infer "paper" quotas. I'm not convinced but I will see what the main text says on this.

Removed over the course of revisions, including from the title.

L32-34 – though of course, this does not mean that these species are threatened by harvest/trade. Why use this 24% figure? Why not use the number of species actually in international trade? The World Wildlife Trade report estimated direct trade involved 12,000 species in a recent 5 year period. Suggest citing this report instead. Scheffers et al. conflate species being in trade with threatened by trade.

We have added the reference to the WWT report and reworded this sentence. Scheffers et al. does not conflate in trade with threatened by trade.

L34-35 – Conversely, if trade is well-managed it can bring myriad benefits to species and people. E.g., for rhinos, it can help support population growth rates. I suggest the authors make this point explicitly in the paragraph.

We make this point in the first sentence of the paper, highlighting trade supports local livelihoods and food security. We have added an additional sentence highlighting the potential benefits.

“Effective management of wildlife trade is crucial in ensuring sustainable trade, supporting local livelihoods, conserving species, and ensuring food security (Booth et al., 2021; Nielsen et al., 2018).”

“While well-managed trade can bolster conservation effects and populations, poorly managed trade can correlate with severe declines in species abundance (Morton et al., 2021).”

L37 – capital L needed?

Removed.

L38-42 – Instead of saying “many”, why not include further detail on quota use. The CITES website has details of quotas for countries and species. I suggest the authors make this change, which will make it easier for the reader to grasp how widespread is the use of quotas.

We have added the sheer number of quotas.

L40 – mandated by who? Do you mean that quotas don’t appear in the Convention text or similar? Please clarify.

By the Convention. Added.

L43 – annual quotas can take the place of multiple NDFs; suggest making this explicit.

Stated in the subsequent sentence.

“As such, scientifically-informed annual quotas can take the place of multiple separate non-detriment findings”

L47-48 – suggest revising syntax. This is an awkward sentence to read.

Reworded.

“Building mechanisms and tools to deliver sustainable trade (e.g. via quotas) is critical and can avoid trade shifting to potentially more damaging illicit pathways (Rivalan et al., 2007).”

L51-52 – effectiveness would be more appropriate here rather than importance, surely?

Edited.

L54 – complied with by whom? Trade chain actors as well as agencies implementing CITES? Suggest making this clear and explicit.

By Parties writ large if they are to be effective. We have revised to make it clearer that we are referring to the national bodies implementing CITES.

“Ensuring export quotas positively contribute to sustainable trade requires three things: 1) quotas represent scientifically-informed sustainable offtakes; 2) quotas are complied with by national agencies implementing CITES; and 3) quotas are adaptive, adjusting through time to reflect new information.”

L57-58 – what do you mean by rigidity here? Parties can set some quotas unilaterally so in theory could be used for many species at scale. Please elaborate on rigidity so it is clear for the reader.

Removed.

L66 – do you mean thousands of individuals? Please clarify.

Yes, now clarified.

L68 – I’m not convinced a key research frontier is understanding quota coverage. The species-country combinations that use quotas can be easily downloaded from the CITES website. Compliance and adaptiveness have some merit as research avenues but see other points on how this has been done.

We have edited the phrasing to reflect our aims more clearly.

“Without a sound understanding of quota coverage, compliance, and adaptiveness globally, the effectiveness and importance of quotas as a management tool cannot be known.”

L89-89 – this should say “Res. Conf. 14.7...”. Please correct.

Corrected.

L105 – I would amend the wording here. Zero-quotas, while having the effect of a ban, are not the same as a ban in CITES terms. They may be changed much more easily and the burden of proof to do so is much lower than would be needed to transfer a species from CITES Appendix I to II for instance.

Edited throughout.

Fig 2E and throughout – As above, I would refrain from referring to zero-quotas as bans. Proxy bans may be appropriate.

Edited throughout.

L104-110 – So, compliance is high – this is a good thing.

L117-118 and Fig. 2G – This is also a good thing; most quotas are not being exceeded.

Agreed, we highlight this high compliance here, in the title, the abstract and the discussion.

Table 1 caption – check syntax.

Corrected

Fig. 3a – It is not clear why the y-axis scale is used. Please explain/justify for the reader.

Explained fully in the Methods. We applied group-level scaling to fit the model and be able to extract meaningful average patterns.

L213-218 – This paragraph focuses on the negative aspects of quota use when one of the key results of this manuscript is that there are high levels of compliance with quotas. I strongly suggest the authors reframe the paragraph to explicitly recognise the high levels of compliance.

Reworded to focus on the high compliance and general increasing coverage for threatened species. This paragraph has been fully rewritten as part of the revisions.

“Our analysis highlights large-scale quota compliance and quantifies the diversity of pathways in which quotas can promote trade in species, prevent increasing trade, and reduce trade below previous levels. Similarly, these results highlight the general decline in the number of poorly understood (Data Deficient or Not Evaluated by the IUCN), globally threatened, and species likely to be specifically threatened by international trade traded outside of quotas.”

L214 – what does vague coverage mean? Please elaborate.

Reworded, and covered in detail in the next paragraph.

L214 – inappropriately high volumes – while this may be the case it is for a small proportion of species only. This should also be stated explicitly.

We have moderated our language substantially here.

“These issues predominantly stem from the opacity of evidence supporting potentially high quotas, quotas remaining unchanged for many years, and ambiguity in quota coverage rather than systematic non-compliance. While quotas have great potential as a key management tool, we urgently need to improve their relevance and effectiveness to ensure international trade is sustainable for such a heavily traded taxon.”

L221-231 – this paragraph epitomises one of the main criticisms I have of this paper. That is, while the authors analyse trends in trade with respect to quota-setting they fail to contend with the reality of trade and make assumptions about the management of species for which quotas are applied. For instance, why did the authors not liaise with key quota-setting countries (ID, MG, ET, GY and SR) to determine the rationale for quota-setting and how decisions related to quotas were made? This needn't have been for all species/quotas but could have focused on countries where non-compliance appeared to be an issue and/or those taxa for which quotas were exceeded the most. This would have provided useful insight to contextualise the other findings. There may or may not have been issues or new information which supported the application of quotas as they have been applied. At the moment this paragraph reads like the authors have looked at the data and just made assumptions about management of quotas, including for the two species examples used, whereas liaison with exporting countries could have provided insight making the dataset richer and the article much more compelling.

The paragraph concerns the publishing of quotas with incomparable terms, not non-compliance as the Reviewer's comment suggests.

The publishing of quotas that combine incomparable terms is a conservation concern regardless of Party as it renders analyses of compliance by Authorities and the Secretariat difficult/impossible thus partially invalidating one of the key points of quotas. We have, however, substantially rewritten this section and conducted additional analyses to highlight the frequency and scale of quotas that use incomparable terms.

*“To be effective, quotas must be complied with. Our results demonstrate this is largely the case, but also highlights the novel risk that a proportion of quotas are ambiguously worded, creating a barrier to effective monitoring of compliance. For example, Indonesia has set ambiguous quotas covering both “skins and skin products” for 15 species, including large quotas for *Crocodylus novaeguineae*, *Varanus salvator*, and *Naja sputatrix* as recently as 2020. For quotas by some Parties, this may be a historical*

issue, e.g. Nicaraguan spectacled caiman (Caiman crocodilus fuscus) quotas aggregated incomparable terms up to 2010 (“skins, sides, tails, bellies, bodies and leather products”) but have since changed this to use comparable terms. Yet the general trend through time appears to suggest increasing numbers of incomparable terms in use rather than less (Figure S7). Likewise, Indonesia exports tens of thousands of live oriental rat snakes annually, despite national quotas stipulating 334-500 live individuals annually (between 2006 and 2023). This occurs because of a separate annual quota for up to 90,000 “skins” or “skins and meat”, the majority of which is traded live and assumed to be destined for processing into skins. While such instances represent a quota-compliant number of individuals in trade, they contribute to the opaqueness of quota regulation, hindering transparent compliance assessments.”

L222 – is this monitoring of trade or species in the wild, or both. Please state explicitly so it is easy for the reader to grasp.

It creates a barrier to monitoring quota compliance. This has repercussions for species in the wild, but at its core it’s a compliance monitoring issue. Clarified.

“To be effective, quotas must be complied with. Our results demonstrate this is largely the case, but also highlights the novel risk that a proportion of quotas are ambiguously worded, creating a barrier to effective monitoring of compliance.”

L243-252 – I don’t doubt that this is an issue but it doesn’t affect many species (35?), which is a very small proportion of CITES-listed species. Again, deeper exploration of the management of such species – through liaison with relevant agencies and stakeholders - could reveal interesting insight into what is happening. This includes where such trade takes place because of genuine corruption or because of other factors e.g., lack of capacity with the relevant CITES authorities – both importers and exporters.

These 36 species are the only species that had publically available NDFs that the Secretariat could assess. If there is a problem with 41% of these 36 NDFs it stands to reason that the quality of the tens of thousands of NDFs that are not publically available follow similar proportions. Thus, this is likely a major issue.

L255-256 – This is not inherently negative for biodiversity where quotas are well evidence-based but the manuscript provides little detailed insight into the extent or nature of management for species subject to quotas.

We have completely rewritten this section and now explicitly say this.

*“For many species (e.g. *Varanus beccarii*, *V. dumerilii*, *V. jobiensis*, and *V. rudicolis* exports from Indonesia), quotas were used to curb high and rising export volumes. However, quotas were most commonly set higher than prior or current trade volumes and often went many years without change. This is not inherently a concern where quotas are merely higher than current patterns of demand and are sustainable (CITES Secretariat, 2013).”*

L255-261 – Again, this is where some investigation into the management of species with quotas could have helped inform the discussion. There are likely many other reasons why quotas may not have changed beyond the 4 reasons the authors suggest. These 4 reasons are also largely unsupported empirically and the manuscript does read like the authors are to a large extent guessing as to why the quotas may not have changed.

We have rewritten this section and removed reference to four specific reasons.

“While our results highlight quotas are largely complied with, have increasing coverage of threatened species, and can address rising trade volumes, our results also highlight potential issues with their use. These issues may present a barrier to sustainability, such as incomparable terms being combined in a single quota and quotas remaining unchanged for many years reflecting a lack of adaptive management.”

L268-270 – See above comments on digging deeper into the management of species for which quotas are set. Such justification may or may not exist for different species and country combinations but we don't know and the authors don't present much evidence in support of their arguments here.

See previous explanation regarding contacting Parties. However, we would like to emphasise that if quota setting and NDF approaches were openly accessible (at least to some degree as we advocate for in our discussion) then there would be no need for such theorising or questioning as the methods could be clearly seen supporting that level of sustainable offtake.

L277-281 – The authors fail to contend with the reality that many CITES authorities are chronically under resourced, which could explain some of the issues with quota setting and a lack of “hard” data and evidence where there are problems with levels at which quotas are set.

This a valid point that we now highlight, but it doesn't change the fact that if there is no funding to produce evidence-based quotas then the end result will be quotas that are not based upon evidence, which is conservation issue. We now address this specifically in a subsequent paragraph.

“The Global Biodiversity Framework 2030 Target 5 focuses on the sustainable use and trade of wild species yet has no listed data indicators to capture the legal wildlife trade (excluding fish stocks), creating such data must be a priority. Such an amendment would be significant, and even if accepted by the majority of Parties, slow to enter into force, but by identifying capacity and implementation gaps in Parties NDF practices, such NDF reporting could be used to leverage increased funding for key Parties from global sources (e.g. the Global Biodiversity Framework Fund).”

L277-281 – the main point made here is not well supported by literature. The authors include 3 references only, two of which are dated and for which circumstances may have changed in the last 20 years. The authors need to support this point much more substantially if they are to retain this in the article.

We respectfully disagree. We have added reference to a recent substantial review (Hughes et al., 2023 *Jrnl Env Man*) that highlights the issue that quotas and NDFs are often based upon poor data. We also reference another recent study that makes the same point for a specific species.

“We echo multiple prior calls for serious international discussion and potential reforms to CITES to more robustly embed an evidence-based framework and justified quotas in national practice (Hughes et al., 2023b, 2023a; Morton et al., 2022). Legally trading wildlife internationally does not equate to sustainable international wildlife trade by default (Trouwborst et al., 2020; Castello & Stewart, 2010; Dumenu, 2019), there must be evidence showing this and this must be publicly available.”

L281-284 – The authors have not done enough to coin this term in my opinion. Many of the arguments used in the manuscript are based on speculation, selected examples, and are not well supported when it comes to understanding how quota-setting relates to the actual management of species by national CITES and other authorities.

This is a repeat of an earlier comment. See response No. 5 above.

L285 – Again, insight into the management of species could reveal interesting insights. Have quotas been set because species are about to be traded and/or countries are preparing for trade in particular species from wild and/or captive sources?

This is something we now highlight in the results that quotas can clearly be seen to be playing both roles – either set to facilitate the (hopefully) sustainable use of species previously not traded or to curb the

growing trade in species where such trade is becoming a concern. Which further highlights the value of quotas, assuming the set volumes are appropriate

“Our analysis highlights large-scale quota compliance and quantifies the diversity of pathways in which quotas can promote trade in species, prevent increasing trade, and reduce trade below previous levels.”

L285-286 – A counter-argument could be that Parties have more up to date information than is currently available publicly and have therefore justified higher quotas, but this isn't captured by the authors because they haven't engaged with the relevant authorities.

A counter-point here is that if such data exists then that is all the more justification for open-access data sharing. More up-to-date information on populations, offtakes etc. has wider relevance beyond trade and CITES, with such data also crucial to addressing the wider biodiversity crisis.

The key point thus is that neither the authors nor the Reviewer can be proven wrong or right at any meaningful scale, because there is no commitment for Parties to share data or methods (of course excluding ranges/coordinates for sensitive species). In the 21st century and given the severity of the global extinction crisis, this is a shocking state of affairs.

L289-291 – It is a shame the authors didn't consider engaging some CITES Parties when conceiving and carrying out this study, because doing so would have enabled them to answer this question, at least in part.

This was removed over the course of revisions but see our previous response regarding engaging Parties and our concerns over whether any information would have been received and its validity.

L294 – What does further discussion mean? By whom, with whom, with what purpose, and in what context? Does this relate to CITES or is this discussion nationally? Or otherwise?

Reworded to be clearer.

“We echo multiple prior calls for serious international discussion and potential reforms to CITES to more robustly embed an evidence-based framework and justified quotas in national practice (Hughes et al., 2023b, 2023a; Morton et al., 2022).”

L295-298 – What evidence? Please be explicit. Acceptable to who? Please be explicit. What is the rationale for publicly available evidence? And, how realistic do the authors consider this to be, both within and outside the CITES context.

Reworded, and removed unclear wording like “acceptable”.

The rationale for publicly available evidence is the same across all science over the last decade: Open data and methods are where trust in the data, analysis, and outputs stem from. There must be evidence that sustainability assessments are carried out at scale and that appropriate methods are being used, correct inference being drawn, etc.

There is clear evidence that the current system and approach to NDFs is inconsistent at best and a clear failure at worst. Evidence for this can be seen in CITES' own documents (AC31 Doc. 14.1/PC25 Doc. 17) where of the handful of NDFs that could be assessed, swathes were missing key data on threats, population trends, correct interpretation of the precautionary principle, data triangulation, and more.

We fully support that sensitive data on rare/desirable species exact distributions should not be available publicly. But the basic data, populations size, trends, view of threats, simulated demographics etc. should be available at the very least to *show based on X, Y and Z data we have concluded that species A has a positive NDF and can be traded in X amount annually by carrying out XYZ methodology*. Fields with far more sensitive data (medicine, clinical psychology etc.) are leading the data revolution and sharing suitably anonymised data to enable replication of their conclusions. Why can conservation science not echo this?

L296-297 – As per a previous comment, lack of an update to a quota is not necessarily cause for conservation concern. The authors need to address this point here and throughout the manuscript. This includes at L304-305; dialogue with key Parties could have been insightful here.

This is a concern also raised by CITES Secretariat in CITES Resolution Conf. 14.7 (Rev. CoP15) specifically in regard to quotas not being updated to reflect changing threats. Thus, as our study highlights, quotas often do not change particularly frequently. This is a concern we should be actively trying to alleviate by requiring evidence justifying quotas (e.g. submit the NDFs supporting your sustainable quota.).

“especially for species threatened by multiple stressors (Symes et al., 2018). This specific issue is identified in CITES Resolution Conf. 14.7 (Rev. CoP15) where the document highlights the concern that quotas may not be updated to reflect changing climatic threats such as droughts (CITES, 2013b).”

“Firstly, establishing and enforcing a minimum acceptable level of information (e.g. 1 or more comparable terms) as potentially ambiguous quotas only further entrench uncertainty, this would likely require a minor revision to the Annex Guidelines of Resolution Conf. 14.7 (Rev CoP15). Secondly, requiring Parties to submit NDF's supporting quotas each time they are communicated to the CITES Secretariat, thus allowing scrutiny of the quota's validity and relevance to current populations and threats.”

L298-301 – Systematic policy change? Which policies and what change are the authors calling for? This reads as an afterthought and it is not at all clear what the authors are suggesting in policy terms.

We have now stated explicitly what we suggest and rewritten this paragraph.

“Firstly, establishing and enforcing a minimum acceptable level of information (e.g. 1 or more comparable terms) as potentially ambiguous quotas only further entrench uncertainty, this would likely require a minor revision to the Annex Guidelines of Resolution Conf. 14.7 (Rev CoP15). Secondly, requiring Parties to submit NDF’s supporting quotas each time they are communicated to the CITES Secretariat, thus allowing scrutiny of the quota’s validity and relevance to current populations and threats.”

L302-309 – So, what changes are actually being proposed? It is not at all clear. For example, are the authors proposing a new CITES Resolution to guide Parties, revision of existing guidance on the use of quotas or something entirely different? The authors need to elaborate on their recommendations so the reader can grasp what is being proposed.

We have now stated explicitly what we suggest and highlighted that one suggestion is much more feasible than the other.

“Firstly, establishing and enforcing a minimum acceptable level of information (e.g. 1 or more comparable terms) as potentially ambiguous quotas only further entrench uncertainty, this would likely require a minor revision to the Annex Guidelines of Resolution Conf. 14.7 (Rev CoP15). Secondly, requiring Parties to submit NDF’s supporting quotas each time they are communicated to the CITES Secretariat, thus allowing scrutiny of the quota’s validity and relevance to current populations and threats.”

L306 – Amendments to what? Importing countries already report trade in CITES-listed species, at least for some species.

Some countries voluntarily report all imported trade. However, as per the Convention, only trade in Appendix I species must be reported by importers and Appendix I species make up a minority of listed species and minuscule proportion of all global trade. So, importer reporting of all trade would be a significant change. However, we no longer make this suggestion here.

L308-309 – In my opinion, the authors cannot claim that this is categorically the case regarding quotas because they failed to ask relevant CITES agencies about their management practices and otherwise only speculate on the reasons for trends in trade of species with established quotas.

In the absence of any public data justifying CITES quotas, it is completely correct and appropriate for us to state that “evidence is sorely lacking”. However, we have reworded this to be clearer.

“Sustainable international trade cannot exist without an evidence-based framework, and such evidence is sorely lacking.”

Reviewer #2

This is a well-written manuscript identifying important issues around the setting and managing of quotas for the international wildlife trade. The study uses empirical evidence to highlight questionable patterns in quota setting and management that deserve more attention and scrutiny. I believe this manuscript merits publication and that a wide audience would be interested in its findings. I have made suggestions mostly for improving clarity and readability of the findings.

Thank you for reviewing our study and providing suggestions for improvement.

-The term “we” is used a few times, consider describing the community / people being referred to here.

We have corrected the use in the Abstract as it was confusing and elsewhere we are referring to ourselves (the authors).

-Global comment: The percentages provided vary throughout the manuscript in terms of how many decimals are used (some are whole numbers, some to the tenth, others to the hundredth); personally I think rounding to whole numbers would be appropriate for this work (in text and in tables/figures).

We have standardised this to show one decimal place, except in Table S1 where for brevity and to ensure rows are readable, we rounded all percentages to whole numbers.

-Lines 15-16; Perhaps omit “sustainable” here in the description of a quota and instead list “sustainability” as an area where we lack sufficient global assessments.

Done, now reads *“Quotas are a common trade management tool, specifying an annual number of individuals to be exported, yet currently there is no global assessments of quota coverage, sustainability, and compliance”*

-The “quota coverage” section answers the “what” quotas cover but does not seem to answer the “where” as specified in line 73.

We have removed this and rewritten the objectives. We cover the “where” in the compliance sections where we outlines where sets quotas and where breaches quotas.

“We have four objectives: (1) assess quota non-compliance; (2) quantify volume changes pre- to post-quota setting; (3) determine whether quotas are consistently updated as per adaptive management; and (4) identify key species and country combinations not utilising quotas.”

Line 78-80 - Somewhere in the beginning of the “quota coverage” section, I suggest having one sentence that defines the key terms you will be referring to without along with examples, specifically for: “term”, “source”, and “purpose” (you list examples for “term” and “source”, but not for “purpose”, I would put all together in an explanatory sentence; i.e. not one with any findings).

This section has been removed during our revisions. We have added purpose examples to the text.

-In the “quota coverage” section you say “term-monitored” (line 80) and on line 86 you say “type of item managed” - I think it would help the reader if you were consistent in your terminology.

This section has been moved to the over the course of revisions. We have added purpose examples to the text.

Corrected in text.

-Similarly, in the text you refer to the “species-source” (line 80-81) and in the figure it only refers to “source” - would be helpful if consistent (maybe just delete species).

Corrected.

Line 88-90: Perhaps provide a bit more explanation here rather than just saying “and presumably referred to to wild trade” with citation.

We detailed the justification for this in the methods and removed it from here over the course of revisions.

-Lines 96-97: In the figure I would provide a quick explanation of the terms used (source, purpose, term). Fig. 1A - Decimal places vary for bar values, I suggest whole numbers or one decimal place. Also, consider consistency in how the bars are labeled, i.e. perhaps “source” goes first in each bar label, then “purpose”, etc. so the reader can more quickly absorb what we are looking at. Not sure why the bars are in the order they are in? Could be helpful to order from highest to lowest or vice versa.

Over the course of revisions this Figure has been moved to the Supplementary Information (Figure S1); nevertheless, we have amended the Figure as suggested.

We retain the order for Fig 1A as it shows increasing quota detail (only source specific to source, term and purpose specific).

Fig. 1B - I think the y-axis should be “Term” to be consistent.

Corrected.

Figure 1 - I am not sure why some of the bars are red? Should mention in the caption. Also, I know the definitions of the various abbreviations are in the supplementary but I think it would help the reader if the terms (or the most commonly used ones if space is an issue?) were also defined in the caption or included in the figures - it is hard to digest with all the unknown acronyms. I would also have the same x-axis for 1b/c/d to enable easy comparisons for the reader.

Corrected.

Line 115 - Perhaps qualify quota breaches as “Recorded” or “Known” quota breaches given there could likely be breaches for which there is no documentation

Added.

Line 118-120 - For this example perhaps provide the number of individuals as well, not just % to give a sense of magnitude.

Added.

Lines 135-136 - The inset panel does not make visible the full distribution (i.e., there is nothing visible after 150% or so...)

The true distribution is shown – it's just at such a low frequency beyond 150% that it is hard/impossible to see, hence the wider main panel focuses on 0-150%.

Table 1 - Suggest spelling out “Exporter-reported” and “Importer-reported” in the top row for ease of reading/understanding figure.

Done.

Line 145 - Says “Using 69 species-party quota time series” - this is a bit unclear, perhaps use plain language to say national-level quotas.

Done.

Lines 152-153 - I would clarify so it is clear you are saying that 46% of quotas were set “above” the pre-quota trade volumes.

Clarified.

Lines 153-155 - Not a complete sentence, need to better explain this point.

Reworded.

“Over 46% (32/69) of quota levels were set above pre-quota trade volumes with no subsequent temporal change over time (‘step change’; Figure 2B). This increases to 62.3% (43/69) of quota series when all quotas representing a step increase and a temporal change are considered (‘step change’ plus relevant ‘step and trend change’; Figure 2B and S4).”

Lines 167-169 - Seems an indirect way of saying that zero quotas (or bans) worked for this species?

Yes, we’re just explicitly phrasing it in terms of the exact trends.

Line 174 - should be “volume” not “volumes”

Corrected.

Lines 176-177 - Caption for “B” is listed twice, combine and make order consistent with caption for “C” (i.e. describe the summary and then the conceptual figures).

Corrected.

Figure 3 - The conceptual figures of how quotas and traded volumes changed after quotas are great (in 3B/3C); I think it would be helpful to label them with letters (e.g. A through R) and then refer to these letters when discussing a specific pattern in the text (e.g., on line 159). I would also consider adding % at the top of each of the bars and possibly even the volumes within each bar.

We have now numbered the concept figures and refer to them in the text and added percentages to the bars.

Line 214-215 - Instead of “vague coverage” perhaps “imprecise data”; I would add “quota” so reads “inappropriately high quota volumes”; and also add quota to “lack of updating quotas”

Done.

Line 224-225 - Delete “is” so reads “rendering compliance impossible”

Corrected.

Line 226 - Should it just be “10,000” - no s? Or instead “10,000-plus”?

Changes to read “tens of thousands...”

Line 227 - Suggest adding “per year” after live individuals

Added annually.

“Likewise, Indonesia exports tens of thousands of live oriental rat snakes annually, despite national quotas stipulating 334-500 live individuals annually (between 2006 and 2023).”

Lines 232-234 - Suggest making first sentence more straightforward, less wordy

Reworded for clarity.

“Using quotas to streamline exporter sustainability assessments by aggregating multiple NDFs requires robust practices that can be scaled up to a Party’s annual export rather than just a population.”

Line 255-256 - perhaps add “set” so reads “most commonly set higher”; instead of “go” should be past tense

Corrected.

“Quotas were most commonly set higher than prior or current trade volumes and often went many years without change.”

Line 261 - You are saying that in most cases it seems the original quota was based on inaccurate data? I would say this there to clarify.

In addressing other Reviewer comments this sentence has been removed.

Line 267 - Suggest deleting “exploited”

Done

Line 277 - You say there is a clear “lack of confidence” is it confidence or instead a “lack of evidence”

Both, but evidence is more quantifiable – corrected.

“Across many taxa and over time, there is a clear lack of evidence that quotas are based on robust data or set sustainably.”

289-291 - Nice

Thank you

298-301 - suggest rewording so sentence starts with “Systemic policy change is necessary to produce

evidence to the contrary.” I am not sure you need the next fragment about an avoidable situation? Could just go right into the changes that must be addressed starting on 302.

Removed as part of revisions-and our suggested changes have been strengthened.

308-309 - great closing sentence

Thank you!

Supplementary line 29: typo of “otherwisree”

Thank you, corrected.

Reviewer #3

The key theme the authors put forward around “Paper Quotas” is an important topic - and the paper makes an important point around the need to scrutinise the export quotas for the international trade in wildlife more systematically in the CITES context to ensure they are fit for purpose. To my knowledge, this is novel and not something that has been looked at systematically in this way before. I do see this topic to be of interest / relevance to those that work on CITES and that are making decisions in the context of international wildlife trade. With overexploitation one of the two key drivers of biodiversity loss, the is a key topic to explore.

Thank you, we agree. Focusing on how we manage species use and trade to ensure sustainability is critically important.

I do wonder, however, if the authors are missing some key aspects of this argument - for instance, they do not look closely at which countries are issuing quotas -- and which are not. Are there populations of globally threatened species that occur in Range States that aren't issuing quotas at all? Or are there countries that are only issuing quotas for only a small number of species e.g. <XX over the last ten years...? The authors should also highlight up front that quotas (in most cases) are voluntary – hence there will be many countries that are not issuing any quotas which may be an even bigger conservation concern than whether the ones that are published are not set at the right level. This “country lens” seems like a key aspect to include – or at least to say that future analyses / focus should look into this aspect in more detail to add further insights into whether CITES quotas are fit for purpose.

We have added additional analyses to unpick what “type” (e.g. high volume, threatened species, etc.) of trade is disproportionately managed or not managed by quotas. See new Figure 4 and the associated results text which highlights countries that trade high numbers of threatened/trade threatened species and currently lack quotas.

Figure 4. Global shifts in trade not managed by national export quotas. A – C. Trends in species richness for species classed as Data Deficient or not assessed by the IUCN (blue tones). D – F. Trends in species richness for species classed as Threatened by the IUCN (Vulnerable, Endangered, Critically Endangered, orange tones). G – I. Trends in species richness for species assessed as likely threatened by international trade (red tones). In A, D and G the dashed lines show the volume of these species traded each year and correspond to the right y-axis.

A key part of the argument is that the quotas are too high -- the evidence around how the authors know they are too high could be strengthened. Just because trade is lower than the quota after it was put in place, doesn't mean the quota is too high -as there are CITES Authorities in each country that are issuing CITES permits and so can control the trade levels and ensure it stays below quota. It also could be that the demand is just not there --so the trade is lower and never reaches the quota level. I'm not saying this is the case -- as there is certainly lots of trade that is unsustainable -- but providing more evidence around the sustainability aspect is important to prove that they are "paper quotas". The paper would benefit from a more detailed look at why the quotas are believed to be unsustainable. Are there specific examples where it is clear that the level of trade is both below the quota and that the trade levels are unsustainable (i.e. the quota is set too high and leading to population declines driven by trade). I'm sure there are many -- but this does not come across clearly in the paper. Think this argument -- which is key - needs more examples. Bringing in the IUCN Red List status and the population trend could help make this case (e.g. some further exploration of globally threatened species with stable or declining

populations where quotas seem inadequate to promote sustainability).

Thank you for this suggestion. Quantifying precisely whether quotas are unsustainable is a critical research frontier for which there remains insufficient (indeed, often no) data available to do so. We have rewrote the section to remove conjecture.

We have also added additional analyses to focus on species that are deemed to be threatened by international trade (Challender et al. 2023) or internationally threatened (as per the IUCN Red List) and highlighted where quotas could be potentially be expanded to cover these species.

Overall, I think this is an important topic and a novel analysis, but I would suggest that it could be strengthened in places to make it more evidence-based and compelling around the need for an overhaul of / more scrutiny on the CITES quota system.

Thank you, we have added a significant new analysis, rethought our recommendations, and corrected a range of minor issues to make a more compelling case.

Detailed comments on specific lines

Abstract:

Lines 20-22: Suggest checking the 62.3% statistic as this seems high – in the section where this is referred to in the text, it mentions Brookesia – and I just wonder if some of this increase relates to quotas for species that may have had a nomenclature change (and so wouldn't have had reported trade previously). Many of the chameleons have split or lumped over the years. This may skew the results?

This is a potential issue, however, we spent considerable time reviewing the taxonomy used in the quota specifications and the UNEP curation process does backdate taxonomic changes in the taxonomies that are accepted at the CoP so don't believe it is the case here.

23 - "were never updated" – this wording is inaccurate in the CITES context-- as technically CITES export quotas are communicated by the CITES Party (government) to the CITES Secretariat and "published" anew on the CITES website (and Species+) each year. It would be more accurate to say "have remained at the same level each year".

Corrected.

"Over 37% of species' quota series remained at the same level each year..."

Introduction:

4031-32 – Suggest adding an explicit mention around conserving the species themselves; this seems to be a gap, but is a key provision of CITES (e.g. ensuring international trade is not detrimental to the species populations in the wild), which you say later.

Added.

“Effective management of wildlife trade is crucial in enabling sustainable use, supporting local livelihoods, conserving species, and ensuring food security (Booth et al., 2021; Nielsen et al., 2018).”

37 – “Listed” isn’t typically capitalised (normally written as “CITES-listed” or “listed in the CITES Appendices”). Could expand to be more clear – e.g. “regulates international trade in the over 40,900 species listed in the CITES Appendices.” Source for total number - <https://cites.org/eng/disc/species.php>

Corrected to “CITES-listed”, here and throughout.

45-46 - As an example -- other reasons may apply e.g. as part of Review of Significant Trade process.

Added.

“Quotas can also be set directly for limited reasons including by the CoP to manage trade when species are moved from Appendix I to II (CITES, 2016), or as part of the Review of Significant Trade process.”

70 – birds - - suggest being a bit careful here - there is still quite a large number of birds traded - internationally and domestically - and large groups (e.g. songbirds) that aren't listed and so there is uncertainty about the scale of this trade. See <https://cites.org/eng/news/technical-workshop-songbird-trade-conservation-management-2023> -- e.g. millions of songbirds in trade...

We agree, and would not wish to understate the risks from the current songbird crisis. We now clarify that listed reptile trade far outweighs trade in other listed vertebrate taxa.

“Unlike certain listed taxa (e.g., birds), the international legal trade in reptiles trade remains vast (>3 million individuals annually), greater than all other listed vertebrate trade combined (Harfoot et al., 2018; Morton et al., 2022), and has the highest number of quotas of any taxon.”

71 – “all other vertebrate trade combined” – needs caveating to say “greater than for all other CITES-listed vertebrate trade combined” (otherwise, the bird issue arises – and perhaps other vertebrates – as per comment above).

Added listed.

79 - Lots removed from original 8470 Species+ download -- when species were removed due to duplication / nomenclature issues - just want to ensure that species that should still be there – e.g. if it has a note that it was originally published for the synonym those have still been retained (e.g.

Chelonoidis carbonarius and Rhampholeon spinosus). (understand that a lot had to go, but was larger than I expected).

This also included duplicated quotas and quotas set that did not cover a standard reporting year but covered specific dates across multiple years, detailed in the quota notes (e.g. many quotas set by Guyana).

79-81 -- As all readers may not be as familiar with quotas, I think it would also be important to state that the species (or taxon) and country and year were also specified in each of these

Added.

“After removing duplication, cleaning, and categorisation, we found 7761 reptile quotas between 1997 and 2023 (each specifying the exporter, year, and taxa), the majority only additionally specified the term monitored (e.g. live, skins, etc.) and the source (e.g. wild, captive, etc., Figure 1A).”

83-84 – Suggest adding summary stat on countries. As countries are so essential to quota setting, it would also be worth adding to total number of countries issuing export quotas - I get 70 (I ignored "Sudan [prior to succession of South Sudan]" to avoid double counting). To me this another important angle -- are there any key reptile range states that are trading and don't issue quotas at all -- or that haven't in the last 5-10 years? Is this a cause for concern? What about the discrepancy between countries -- you'll note that some of the countries that have had sustainability issues flagged in the context of the Review of Significant Trade process have large numbers of quotas -- are there others with fewer published that are actively trading Endangered / Critically Endangered species that don't regularly issue quotas (e.g. have <20)? Note also if you sort by countries, the top five countries represent more than half of the reptile quotas --seems notable... (links to the Review of Significant Trade again...).

See response to point below.

90 – I think the purpose of the trade is not specified in 93.8% cases because the purpose of the trade doesn't really matter in practice in this context – the assumption is that it's for all purposes (e.g. a live animal traded for commercial purposes or a zoo is still a live animal that has been taken from the wild (assuming source is wild), so the trade would count towards the quota). I also suspect that the purpose is not requested by CITES – so the 6.2% that provide a purpose are likely to be very specific purposes... Main point being, I wouldn't make too much of this finding as not sure it has a huge impact on conservation / wildlife management...suggest removing the “purpose” aspect of the analysis or at least downplaying it as I do not see this as crucial to making quotas effective in CITES.

We have moved this Figure to the supplement and shortened this section of text in the manuscript, since on reflection its conservation relevance is relatively low.

In its place we have added a comprehensive analysis of what species and country combinations are currently not covered by CITES quotas, but are of high volume, internationally threatened or assessed to be threatened by international trade. This is reflected additionally in the slightly restructured aims of the paper at the end of the Introduction. Thank you for your suggestion in this regard.

Figure 1 – as per above – suggest removing “Purpose” considerations. Also – year range says to 2021, but elsewhere you say 2023.

That is correct, when looking solely at quotas we used all available years 1997 – 2023. However, when we look at compliance and trends using CITES trade data and quotas, we limit analyses to 1997 to 2021 as it is likely that 2022/23 are not yet fully reported in the data and will be subject to change when the next version of the database is released. (https://trade.cites.org/cites_trade_guidelines/en-CITES_Trade_Database_Guide.pdf, page 2)

Table 1 – suggest putting “potential breaches” as this trade was only over quota on the importer side (and there are reasons why there could be discrepancies). Also – consider putting the IUCN Red List category in this table – were any of these EN / CR species? Add key for what “ER” and “IR” are – or add “ER” and “IR” into table header.

Added.

122- species with quotas for which there wasn't any trade? (rather than quotas with no trade...)

Edited.

“On average, only 49.8% of the maximum permitted offtake under quotas was traded (e.g. only 498 individuals traded out of a quota for 1000); this increased to 69.0% after removing 717 quotas for which there was not any reported trade (Figure 2G).”

155-157 – Not sure about the sentence about Brookesia – was the low trade pre-quota due to nomenclature changes with the genus – or even the Review of Significant Trade Process? There has been a lot of taxonomic revision in this family. Perhaps worth a check before inferring that it may be a success story.

This is not due to nomenclature changes and as far as we can deduce is not to do with RST (<https://rst.cites.org/public/cases>). We have, however, reworded our framing to highlight the uncertainty around this.

Figure 3 – I found this figure a bit challenging to interpret. To me, section "A" (top graph) seems quite

good from a conservation perspective if I'm interpreting it properly -- looks like the trade trend is upward "pre-quota", but then subsequent to the quota setting, the trade stays largely lower than the quota and starts to decrease? Though this is contrary to the overall message of the paper... Also the Figure header has two "B"s and two "C"s" – which I understand is likely top figure and then the part of the figure below - consider revising to ensure it is clear.

Corrected the legend.

186 – 190 - Each quota is communicated to the Secretariat / published anew each year (even if it maintains at the same level). The way this is currently characterised seems a bit misleading...

Edited out wording throughout to remove "update" as the term of choice and instead use quota volumes changing or not.

191-192 - How many of these are COP approved quotas -- including zero quotas - which would make sense for them to stagnate as they are agreed via the CoP...

We agree, but this is part of what we consider to be the problem: 3-years between review and centralised setting is at odds with a system that can respond adaptively to threats species face especially in the era of a rapidly changing world.

All quota series that were composed partly or wholly of zero quotas were removed (see Method text).

Discussion

213- As there are different types of quotas – e.g. harvest quotas, hunting quotas, and export quotas, etc... - suggest tightening up the language so it's clear they are "export quotas" throughout. For instance the first sentence of the Discussion would be more clear if you say "international export quotas published for CITES-listed reptiles."

Edited.

"Our analysis highlights several potential issues with international export quotas published for CITES-listed reptiles."

214 – one of the main points is around "inappropriately high volumes," – but in the section that follows ("Quotas as a tool for sustainable use") – I don't think this argument is effectively made. Which species are at risk from high quotas? Which species have trade levels that are too high and leading to population declines? Which ones are globally threatened, with trade as an on-going threat where the quota is set too high. These seem like important angles for this argument, but not sure the Discussion adequately covers this.

This section has been fully rewritten and no longer makes this argument.

218 – Similarly, tightening up the language around “international trade” (rather than just trade) is also important throughout – e.g. the last sentence should be “to ensure international trade is sustainable in ...”. I’d suggest rephrasing that way (rather than “sustainable use”) as in many cases “ensuring sustainable use” is not the goal (e.g. for species where zero quotas are needed), but rather ensuring that international trade is sustainable. Check throughout (e.g. header of next section).

Edited throughout the text.

223-226 - Are there other, more current examples? This Nicaraguan spectacled caiman example is a bit old at this point as the last time a Nicaragua issued a quota for this was back in 2010 -- since then, it has been "skins, wild taken" - which is pretty clear -- so things have improved!

We have added a Figure to the supplement (Figure S7) showing that the number and proportion of quotas containing incomparable terms is increasing through time unfortunately rather than improving.

We have changed the example we give in the text to highlight more recent and higher volume examples of this.

*“For example, Indonesia has set ambiguous quotas covering both “skins and skin products” for 15 species, including large quotas for *Crocodylus novaeguineae*, *Varanus salvator*, *Naja sputatrix* as recently as 2020. For quotas by some Parties, this may be a historical issue, e.g. Nicaraguan spectacled caiman (*Caiman crocodilus fuscus*) quotas aggregated incomparable terms up to 2010 (“skins, sides, tails, bellies, bodies and leather products”) but have since changed this to use comparable terms. Yet the general trend through time appears to suggest increasing numbers of incomparable terms in use rather than less (Figure S7).”*

242- Perhaps expand further beyond just “then quotas just...”-- e.g. quotas without a firm scientific basis...compound the issue... or “inappropriately set quotas” (e.g. those that are too high?... Think it's worth being explicit to get your point across.

Added.

249 – as there is some uncertainty – suggest reframing “breached” to “apparently exceeded” (see next comment)

Added, “apparent”.

251 - That “potentially” breach (as well as being “potentially illegal”) -- suggest being a bit careful here as there are reasons why discrepancies may occur in the CITES trade data (e.g. year-end trade, importers reporting on the basis of “permits issued” and trade may have been cancelled), etc... so it may not be a breach. See page 14 of the CITES Trade Database Guidance on potential reasons for discrepancies

between importer/exporter parties -- and add to the References if you include (citation is specified in the doc). https://trade.cites.org/cites_trade_guidelines/en-CITES_Trade_Database_Guide.pdf.

Added.

255-267 – In general it's a good thing that quotas are higher than current trade volumes – that could be interpreted that CITES is working – countries are implementing CITES well as they are keeping their trade under the quote. In terms of the scenarios – I think there needs to be some recognition that one scenario is that CITES Parties are keeping trade levels at or below their published quotas rather than demand necessarily declining. In other words, it's important to consider that CITES Authority will stop issuing permits for trade when they reach their quota (ideally!). For scenario 2) not sure I fully understand why other threats would make the quota irrelevant? Explain this more -- are you saying other threats have made the species population decline and that's why they're not being traded? Not sure... For scenario 4 – many species will lack data, so if you're expecting perfect data for the thousands of species, this is unrealistic – I don't think that means the quota is irrelevant (this is where adaptive management would come in...?). Important to strengthen your argument around the sustainability concerns if you want to say this.

We have reworded this section substantially now and now longer use these four scenarios as they were a potential oversimplification.

“Across many taxa and over time, there is a clear concern that quotas are not always based on robust data or set sustainably (Dumenu, 2019. Hughes et al. 2023). Instead, some authors suggest that quotas can be based entirely on subjective information (concerning Aquilaria quotas; Newton & Soehartono, 2001), guesswork rather than hard data (concerning Indonesia's quotas; Soehartono & Mardiasuti, 2002), or at odds with the principles of sustainable use (for leopard quotas; Trouwborst et al., 2020). While our results highlight quotas are largely complied with, have increasing coverage of threatened species, and can address rising trade volumes, our results also highlight potential issues with their use. These issues may present a barrier to sustainability, such as incomparable terms being combined in a single quota and quotas remaining unchanged for many years reflecting a lack of adaptive management.”

285 – A bit worried about criteria 1. Sometimes there are cases where it may be precautionary to have a quota even in the absence of trade. For example – a zero quota. Are any of the 59 zero quotas?

These criteria are no longer referred to. The 58 quotas references are all non zero-quotas.

Conclusions

295 – suggest addition of another paper beyond the lead author on this paper

Added reference to a substantial review paper by a large number of researchers in this field (Hughes et al. 2023 Journal of Environmental Management).

295-296 - Strangely phrased (“it cannot be acceptable...”). Would be stronger if the paper had put forward compelling examples of where the legal trade is unsustainable – and there are many! Then the Conclusion could more clearly draw out the flaws in the current system, etc...

Rephrased to be clearer.

“National authorities should not equate legally internationally traded wildlife with sustainable international wildlife trade, there must be evidence showing this and this must be publicly available”

299-301 – I don’t think this sentence makes sense “The onus must be on systematic policy change to produce evidence to the contrary – a completely avoidable situation if proactively evidencing sustainability had truly been the status quo rather than reactively highlighting unsustainability.” [Also be careful as this paper could be characterised as "reactively highlighting unsustainability"]

Rephrased. We accept the point that this paper, and really all research papers on trade, are either retrospectively highlighting unsustainable or sustainable trade – this is wholly unavoidable. As we highlight in this article, the only way to make this proactive is to embed data collection and data sharing in trade policy – e.g. for internationally traded species at risk from trade through CITES.

“We echo multiple prior calls for serious international discussion and potential reforms to CITES to more robustly embed an evidence-based framework and justified quotas in national practice (Hughes et al., 2023b, 2023a; Morton et al., 2022). Legally trading wildlife internationally does not equate to sustainable international wildlife trade by default (Trouwborst et al., 2020; Castello & Stewart, 2010; Dumenu, 2019), there must be evidence showing this and this must be publicly available.

Systematic policy changes up to and including amendments to the Convention are needed to proactively support the evidenced sustainability of trade. We suggest the following amendments would improve assessing compliance and confidence in sustainable offtake.”

302-304 - Not sure the “vagueness” of the quotas is really an issue in practice (particularly on “purpose” as highlighted above) – and I do think that quotas are getting better defined in recent years (see caiman example I mentioned above) -- not sure the vagueness is really the heart of the problem here. That said, if authors disagree, they could engage directly with the CITES process and suggest changes? Seems like the main issue is ensuring that the quota is set at a scientifically justifiable level that means that trade is not having a detrimental effect on the species in the wild – and where there are declines, the quota should be revisited to determine if a lower quota is needed. This to me, is the most important and

compelling argument to focus on. Suggest considering strengthening this conclusion more (currently missing).

Our added supplementary analysis, unfortunately, highlights that while vague quotas are a small percent (and absolute number) of all quotas, their incidence is growing rather than becoming less prevalent with time.

We have removed the emphasis on specifying the source and purpose in quotas as it is less crucial than guaranteeing the quota's sustainability.

Rephrased thus:

“We suggest the following amendments would improve assessing compliance and confidence in sustainable offtake. Firstly, establishing and enforcing a minimum acceptable level of information (e.g. 1 or more comparable terms) as potentially ambiguous quotas only further entrench uncertainty, this would likely require a minor revision to the Annex Guidelines of Resolution Conf. 14.7 (Rev CoP15). Secondly, requiring Parties to submit NDF’s supporting quotas each time they are communicated to the CITES Secretariat, thus allowing scrutiny of the quota’s validity and relevance to current populations and threats.”

306-307 – importers do already report on Appendix I-listed species. Might just want to specify “all CITES-listed trade”.

Clarified.

308 – not sure this makes sense “to add onus to importer responsibility, rather than just exporters.”

We have rewritten this section to more clearly highlight the changes we suggest.

309 – Same point as above on last sentence –(Technically, sustainable trade CAN exist without evidence...). Ending could be strengthened. Something that really drive home that as the world trades more and more species – and with biodiversity in sharp decline – a stronger and more evidence-based framework for international quota setting and compliance is needed in order to really ensure international trade is sustainable...

We have reworded this to make a stronger more explicit call for an evidence-based system.

367-368 - Fine to remove these and call them duplicates for your purposes, but it's incorrect to call it a mistake - Species+ is set up to show the history and inform CITES implementation, so these are presenting the facts around the quota issuance and provides CITES Parties with important context around CITES nomenclature and historic quotas, etc. (e.g. that a species would have been subject to the quota prior to the split / lump / taxonomic change). Please revise “mistakenly back-dated for new taxa

prior to their accepted existence by the CoP.” Technically, the species did exist, they were just called something else (and traded under that other name).

We have removed mistakenly.

376-379 - why don't you use the RL status or Appendix listing in the analysis? (sorry if I missed it?)

Removed reference to species Appendix listing and we now use species IUCN Red List assessments and likelihood of international trade posing a threat in Figure 4.

386 – 2021 vs. 2023 – think you say 2023 in the manuscript in places – needs check.

We use two different time periods. To describe quotas we use up to 2023 (all published quotas), but to assess compliance we only use those up to 2021 as the CITES Trade Database can't be reliably used up to as recent a date as quotas are available for.

390 – Please refer to this as the CITES Trade Database (not the “CITES database”)

Added.

391-397 – Not sure this is clear why they were removed – quotas can be published under different terms and you could still compare the trade in live with the quota for live...

This was because of an issue we highlight in the Methods – certain country-species combinations have separate quotas for multiple terms of the same species, e.g. Javan spitting cobras from Indonesia which have low quota volumes for “live” trade and high quota volumes for “skins and skin products” or “consumption”. Yet the majority of trade destined for skins is traded live and then subsequently slaughtered for their skins (TRAFFIC, 2008) (rather than traded directly as skins as the quota note implies). We initially included these quotas and compared and contrasted the quotas and actual volumes, but where tens of thousands of live individuals are traded when the quota is only hundreds it appears that the quotas have been breached by >1000%. Hence, we investigated and found reference to the issue in an older TRAFFIC report that deduced most of the export is likely destined for skins (but exported live).

We were in two minds whether to include such instances. On one hand, this is a prime example of vague quota coverage and/or enforcement, as Indonesia may export a quota-compliant total volume of snakes albeit in the wrong form specified by their quotas, so potentially less a sustainability issue and more an administrative/bureaucratic issue. On the other hand, we did not want to highlight certain countries in major breach of national quotas, when we likely know the explanation and its not a sustainability issue.

Hence, we elected to remove all quotas where different terms had independent quotas in the same year (we kept quotas where they were combined e.g. 1000 live or whole skins).

398 - Please add in the name of the database – the CITES Trade Database – along with the link. There are instructions on how to cite in the CITES Trade Database Guidance.

Added.

442 - Unclear why these are removed -- won't this miss out on lots of species with sustainability issues - e.g. those that have gone through the Review of Significant Trade Process and then have CoP-approved quotas?

Potentially, but including them risks negatively biasing the results, e.g. long-term zero quotas may by default always remain zero as the species just cannot be sustainably traded.

444-446 – As in section 255-267, think you need more evidence around them being “irrelevant” --- Compliance could also be high because these are voluntary quotas issued by the countries and there are CITES Authorities managing national trade levels, issuing permits and ensuring trade stays within their national quotas...

This has been removed.

Decision Letter, first revision:

Our ref: NATECOLEVOL-23112755A

17th July 2024

Dear Dr. Morton,

Thank you for your patience as we've prepared the guidelines for final submission of your Nature Ecology & Evolution manuscript, "High compliance and coverage but insufficient adaptive setting in international wildlife trade quotas" (NATECOLEVOL-23112755A). Please carefully follow the step-by-step instructions provided in the attached file, and add a response in each row of the table to indicate the changes that you have made. Please also check and comment on any additional marked-up edits we have proposed within the text. Ensuring that each point is addressed will help to ensure that your

50revised manuscript can be swiftly handed over to our production team.

****We would like to start working on your revised paper, with all of the requested files and forms, as soon as possible (preferably within two weeks). Please get in contact with us immediately if you anticipate it taking more than two weeks to submit these revised files.****

In recognition of the time and expertise our reviewers provide to Nature Ecology & Evolution's editorial process, we would like to formally acknowledge their contribution to the external peer review of your manuscript entitled "High compliance and coverage but insufficient adaptive setting in international wildlife trade quotas". For those reviewers who give their assent, we will be publishing their names alongside the published article.

Nature Ecology & Evolution offers a Transparent Peer Review option for new original research manuscripts submitted after December 1st, 2019. As part of this initiative, we encourage our authors to support increased transparency into the peer review process by agreeing to have the reviewer comments, author rebuttal letters, and editorial decision letters published as a Supplementary item. When you submit your final files please clearly state in your cover letter whether or not you would like to participate in this initiative. Please note that failure to state your preference will result in delays in accepting your manuscript for publication.

Cover suggestions

We welcome submissions of artwork for consideration for our cover. For more information, please see our guide for cover artwork.

Nature Ecology & Evolution has now transitioned to a unified Rights Collection system which will allow our Author Services team to quickly and easily collect the rights and permissions required to publish your work. Approximately 10 days after your paper is formally accepted, you will receive an email in providing you with a link to complete the grant of rights. If your paper is eligible for Open Access, our Author Services team will also be in touch regarding any additional information that may be required to arrange payment for your article.

Please note that *Nature Ecology & Evolution* is a Transformative Journal (TJ). Authors may publish their research with us through the traditional subscription access route or make their paper immediately open access through payment of an article-processing charge (APC). Authors will not be required to make a final decision about access to their article until it has been accepted. Find out more about Transformative Journals

Authors may need to take specific actions to achieve compliance with funder and institutional open access mandates. If your research is supported by a funder that requires immediate open access (e.g. according to Plan S principles) then you should select the gold OA route, and we will direct you to the compliant route where possible. For authors selecting the subscription publication route, the journal's standard licensing terms will need to be accepted, including <https://www.nature.com/nature-portfolio/editorial-policies/self-archiving-and-license-to-publish>. Those licensing terms will supersede any other terms that the author or any third party may assert apply to any version of the manuscript.

[REDACTED]

[REDACTED]

Reviewer #1:

Remarks to the Author:

I have re-read the manuscript in full and the response to the reviewers comments. I commend the authors for the substantial changes they have made to the manuscript, which reflects edits and changes suggested by all reviewers.

However, I'm left with a major concern. The authors call for greater specificity in exactly what quotas cover, justification for unchanged quotas, and transparency over quota determination to ensure high compliance equates to sustainable use. Yet, a major weakness of the study is that the authors fail to try and understand in real world terms why quotas have or have not changed. This could have been achieved by contacting the relevant CITES agencies – contact details for all such agencies are available online. As a result, the authors can only really speculate as to why quotas are being managed in the way that they are. There are few breaches, which could have been explained thorough correspondence with relevant parties.

The authors report that most quotas for live reptiles were set by Indonesia (567), Madagascar (563), Ethiopia (272), Guyana (262), and Suriname (193), and that known quota breaches occurred in 6.3% of quotas, with Madagascar having the most instances (82 – 14% of their set quotas), followed by Ghana (24 – 23% of their set quotas) and Indonesia (21 – 4% of their set quotas). They also note that trade was generally compliant, while the distribution of percentages of quota allowances used is highly right-skewed with a small number of substantial breaches. It seem strange that authors did not reach out to try and understand what the reasons behind these quotas breaches may have been. Given the number of countries involved, gaining an insight into how quotas are managed nationally – plus anything available on species they discuss – would have allowed the authors to discuss solutions in a more meaningful way. In theory, there could be many reasons for such apparent breaches. The same goes for the top 10 breaches (included in Table 1). There is likely a rational explanation for why there are discrepancies between importer and exporter reported data.

Reviewer #2:

Remarks to the Author:

My concerns have been addressed, and I think this improved manuscript will make an important contribution to the literature. My only lingering concern is regarding the title; as it is currently written is

53not so clear to me. Perhaps something like "High compliance and coverage but a lack of adaptive management in setting international wildlife trade quotas".

Reviewer #3:

Remarks to the Author:

Thank you to the authors for their substantial updates to the manuscript and taking on board the first round of very thorough comments from the reviewers. I am happy that my comments have been taken on board and addressed fully. I particularly appreciate the removal of Figure 1 and the inclusion of the additional analysis (Figure 4); this has improved the overall manuscript.

[Note, there are now currently two "Figure 2"'s - so a check there is needed].

No further comments from my perspective. I am supportive of this manuscript being published.

Author Rebuttal, first revision:

Reviewer #1 (Remarks to the Author):

I have re-read the manuscript in full and the response to the reviewers comments. I commend the authors for the substantial changes they have made to the manuscript, which reflects edits and changes suggested by all reviewers.

Thank you.

However, I'm left with a major concern. The authors call for greater specificity in exactly what quotas cover, justification for unchanged quotas, and transparency over quota determination to ensure high compliance equates to sustainable use. Yet, a major weakness of the study is that the authors fail to try and understand in real world terms why quotas have or have not changed. This could have been achieved by contacting the relevant CITES agencies – contact details for all such agencies are available online. As a result, the authors can only really speculate as to why quotas are being managed in the way that they are. There are few breaches, which could have been explained through correspondence with relevant parties. The authors report that most quotas for live reptiles were set by Indonesia (567), Madagascar (563), Ethiopia (272), Guyana (262), and Suriname (193), and that known quota breaches occurred in 6.3% of quotas, with Madagascar having the most instances (82 – 14% of their set quotas), followed by Ghana (24 – 23% of their set quotas) and Indonesia (21 – 4% of their set quotas). They also note that

54trade was generally compliant, while the distribution of percentages of quota allowances used is highly right-skewed with a small number of substantial breaches. It seem strange that authors did not reach out to try and understand what the reasons behind these quotas breaches may have been. Given the number of countries involved, gaining an insight into how quotas are managed nationally – plus anything available on species they discuss – would have allowed the authors to discuss solutions in a more meaningful way. In theory, there could be many reasons for such apparent breaches.

We acknowledge the point being made and do agree that what the reviewer suggests would make an interesting and very worthwhile study. However, in line with the editor we also agree that the suggested analysis represents a separate and very different piece of research.

We also remain concerned (based on prior experience as highlighted in our previous response) that eliciting the information from management authorities is not as simple as the reviewer suggests and more likely to be ignored than answered. Similarly, we remain concerned that even if such queries were answered how we could triangulate the accuracy of any reasons we received.

In line with the editor’s recommendation, we have added text to the discussion highlighting how working with/following up individual species-Party combinations is an option but doing this at scale will quickly become unmanageable and the issue would be easier mitigated now and into the future with more transparent reporting.

“Liaising with specific Parties and species could offer insight into quota setting practices, sustainability and any reasons for breaches, but at a global scale for hundreds of species-party combinations this will quickly become unmanageable. Open and transparent methods and justification for quotas are needed to ensure sustainability at scale.”

The same goes for the top 10 breaches (included in Table 1). There is likely a rational explanation for why there are discrepancies between importer and exporter reported data.

We highlight these as being of interest as reasons for discrepancies typically are expected to be the opposite e.g. exporter reported being greater than importer reported (as importers are not required to report App II, and exporters may report permitted amounts rather than actual amounts – despite this being discouraged).

We have added a brief section highlighting this in the methods.

“We also highlight records where exporter-reported volumes are quota compliant yet importer-reported volumes breach quotas (Table 1 and S2). Such discrepancies could potentially be used to highlight additional breaches or reporting errors and inconsistencies, although due to differing reporting practices between Parties exporter and importer reported data can vary.”

Reviewer #2 (Remarks to the Author):

My concerns have been addressed, and I think this improved manuscript will make an important contribution to the literature. My only lingering concern is regarding the title; as it is currently written is not so clear to me. Perhaps something like "High compliance and coverage but a lack of adaptive management in setting international wildlife trade quotas".

We thank the reviewer for their constructive previous comments and are happy they find the revised manuscript a valuable contribution.

We have reworded the title to instead read:

"International wildlife trade quotas are characterised by high compliance and coverage but insufficient adaptive management"

Reviewer #3 (Remarks to the Author):

Thank you to the authors for their substantial updates to the manuscript and taking on board the first round of very thorough comments from the reviewers. I am happy that my comments have been taken on board and addressed fully. I particularly appreciate the removal of Figure 1 and the inclusion of the additional analysis (Figure 4); this has improved the overall manuscript.

Thank you again for the helpful previous round of comments we were happy to make the changes and believe the manuscript is much improved by their addition.

[Note, there are now currently two "Figure 2"s - so a check there is needed].

We have corrected this now.

No further comments from my perspective. I am supportive of this manuscript being published.

Thank you, we've appreciated the comments.

Final Decision Letter:

30th July 2024

Dear Dr Morton,

We are pleased to inform you that your Article entitled "International wildlife trade quotas are characterised by high compliance and coverage but insufficient adaptive management", has now been accepted for publication in *Nature Ecology & Evolution*.

Over the next few weeks, your paper will be copyedited to ensure that it conforms to *Nature Ecology and Evolution* style. Once your paper is typeset, you will receive an email with a link to choose the appropriate publishing options for your paper and our Author Services team will be in touch regarding any additional information that may be required

Due to the importance of these deadlines, we ask you please us know now whether you will be difficult to contact over the next month. If this is the case, we ask you provide us with the contact information (email, phone and fax) of someone who will be able to check the proofs on your behalf, and who will be available to address any last-minute problems . Once your paper has been scheduled for online publication, the Nature press office will be in touch to confirm the details.

Acceptance of your manuscript is conditional on all authors' agreement with our publication policies (see www.nature.com/authors/policies/index.html). In particular your manuscript must not be published elsewhere and there must be no announcement of the work to any media outlet until the publication date (the day on which it is uploaded onto our web site).

Please note that *Nature Ecology & Evolution* is a Transformative Journal (TJ). Authors may publish their research with us through the traditional subscription access route or make their paper immediately open access through payment of an article-processing charge (APC). Authors will not be required to make a final decision about access to their article until it has been accepted. Find out more about Transformative Journals

Authors may need to take specific actions to achieve compliance with funder and institutional open access mandates. If your research is supported by a funder that requires immediate open access (e.g.

57according to Plan S principles) then you should select the gold OA route, and we will direct you to the compliant route where possible. For authors selecting the subscription publication route, the journal's standard licensing terms will need to be accepted, including <https://www.nature.com/nature-portfolio/editorial-policies/self-archiving-and-license-to-publish>. Those licensing terms will supersede any other terms that the author or any third party may assert apply to any version of the manuscript.

We welcome the submission of potential cover material (including a short caption of around 40 words) related to your manuscript; suggestions should be sent to Nature Ecology & Evolution as electronic files (the image should be 300 dpi at 210 x 297 mm in either TIFF or JPEG format). Please note that such pictures should be selected more for their aesthetic appeal than for their scientific content, and that colour images work better than black and white or grayscale images. Please do not try to design a cover with the Nature Ecology & Evolution logo etc., and please do not submit composites of images related to your work. I am sure you will understand that we cannot make any promise as to whether any of your suggestions might be selected for the cover of the journal.

You can generate the link yourself when you receive your article DOI by entering it

here: <http://authors.springernature.com/share>.

Thank you for choosing NEE to publish your work; I look forward to seeing it published soon.

[REDACTED]

P.S. Click on the following link if you would like to recommend Nature Ecology & Evolution to your librarian <http://www.nature.com/subscriptions/recommend.html#forms>

** Visit the Springer Nature Editorial and Publishing website at www.springernature.com/editorial-and-publishing-jobs for more information about our career opportunities. If you have any questions please click here.**